# MaskBit: Embedding-free Image Generation via Bit Tokens

**Mark Weber**                                                                      *mark-cs.weber@tum.de*
*Technical University of Munich, MCML*

**Lijun Yu**
*Carnegie Mellon University*

**Qihang Yu**
*ByteDance*

**Xueqing Deng**
*ByteDance*

**Xiaohui Shen**
*ByteDance*

**Daniel Cremers**
*Technical University of Munich, MCML*

**Liang-Chieh Chen**
*ByteDance*

**Reviewed on OpenReview:** *https://openreview.net/forum?id=NYe2JuN3v3*

## Abstract

Masked transformer models for class-conditional image generation have become a compelling alternative to diffusion models. Typically comprising two stages – an initial VQGAN model for transitioning between latent space and image space, and a subsequent Transformer model for image generation within latent space – these frameworks offer promising avenues for image synthesis. In this study, we present two primary contributions: Firstly, an empirical and systematic examination of VQGANs, leading to a modernized VQGAN. Secondly, a novel embedding-free generation network operating directly on bit tokens – a binary quantized representation of tokens with rich semantics. The first contribution furnishes a transparent, reproducible, and high-performing VQGAN model, enhancing accessibility and matching the performance of current state-of-the-art methods while revealing previously undisclosed details. The second contribution demonstrates that embedding-free image generation using bit tokens achieves a new state-of-the-art FID of 1.52 on the ImageNet $256 \times 256$ benchmark, with a compact generator model of mere 305M parameters. The code for this project is available on `https://github.com/markweberdev/maskbit`.

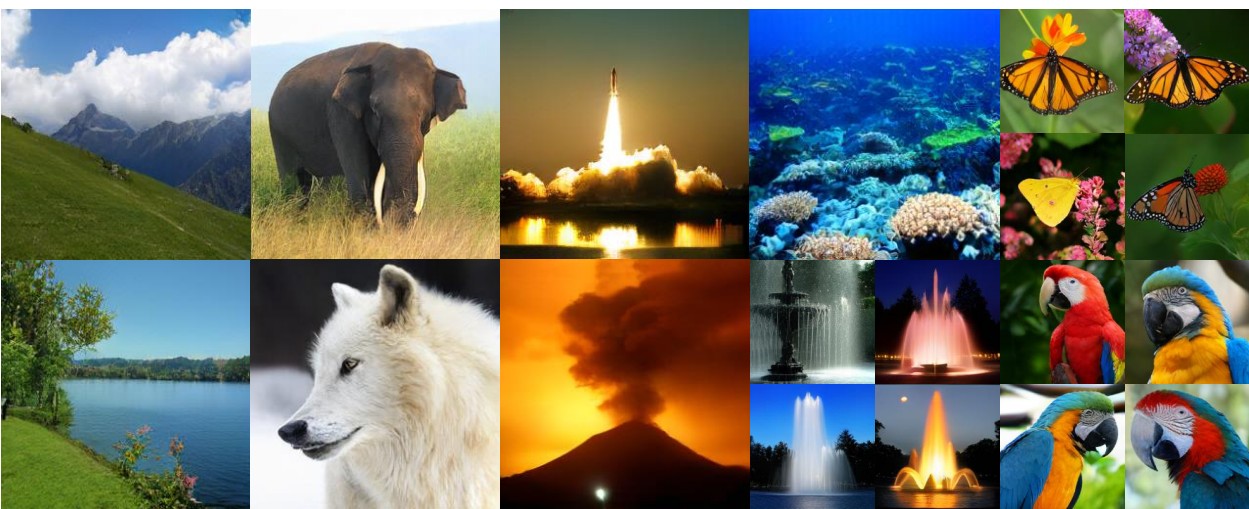

Figure 1: **Generated images by the proposed MaskBit.** We showcase samples from MaskBit trained on ImageNet at $256 \times 256$ resolution.

# 1 Motivation

Masked transformer models for class-conditioned image (Chang et al., 2022; Yu et al., 2024a) or text-to-image (Chang et al., 2023) generation have emerged as strong alternatives to auto-regressive models (Esser et al., 2021; Yu et al., 2022a; Lee et al., 2022a) and diffusion models (Ho et al., 2020; Rombach et al., 2022). These masked transformer methods typically employ a two-stage framework: a *discrete tokenizer* (*e.g.*, VQGAN (Esser et al., 2021)) projects the input from image space to a discrete, compact latent space, while a transformer model (Vaswani et al., 2017; Devlin et al., 2018) serves as the generator to create images in latent space from a masked token sequence. These methods achieve performance comparable to state-of-the-art auto-regressive and diffusion models, but with the advantage of requiring significantly fewer sampling steps and smaller model sizes, resulting in substantial speed-ups.

Despite the success of masked transformer frameworks, the development details of a strong tokenizer have been largely overlooked. Moreover, one key aspect of training modern VQGAN-based tokenizers (Chang et al., 2022; Yu et al., 2023; 2024a) – the perceptual loss – remains undiscussed. Since latent space-based generation relies on encoding and decoding the input image with the VQGAN tokenizer, the final image quality is influenced by both the generator network and the VQGAN. The importance of a publicly available, high-performance VQGAN model cannot be overstated, yet the most widely-used model and code-base still originate from the original work (Esser et al., 2021) developed over three years ago. Although stronger, closed-source VQGAN variants exist (Chang et al., 2022; Yu et al., 2023; 2024a), their details are not fully shared in the literature, creating a significant performance gap for researchers without access to these advanced tokenizers. While the community has made attempts (Besnier & Chen, 2023; Luo et al., 2024) to reproduce these works, none have matched the performance (for both reconstruction and generation) reported in (Chang et al., 2022; Yu et al., 2023; 2024a).

In this paper, we undertake a systematic step-by-step study to elucidate the architectural design and training process necessary to create a modernized VQGAN model, referred to as VQGAN+. We provide a detailed ablation of key components in the VQGAN design, and propose several changes to them, including model and discriminator architecture, perceptual loss, and training recipe. As a result, we significantly enhance the VQGAN model, reducing the reconstruction FID from 7.94 (Esser et al., 2021) to 1.66, marking an impressive improvement of 6.28. Moreover, our modernized VQGAN+ improves the generation performance and establishes a strong, reproducible foundation for future generation frameworks based on VQGAN.

Furthermore, we delve into embedding-free tokenization (Yu et al., 2024a; Mentzer et al., 2024), specifically implementing Lookup-Free Quantization (LFQ) (Yu et al., 2024a) within our improved tokenizer framework.

Our resulting method, employs a binary quantization process by projecting latent embeddings into $K$ dimensions and then quantizing them based on their sign values. This process produces bit tokens, where each token is represented by $K$ bits. We empirically observe that this representation captures high-level structured information, with bit tokens in close proximity being semantically similar. This insight leads us to propose a novel embedding-free generation model, MaskBit, which directly generates images using *bit tokens*, eliminating the need for learning new embeddings (from VQGAN token indices to new embedding values) as required in traditional VQGAN-based generators (Esser et al., 2021; Chang et al., 2022; Yu et al., 2024a). Consequently, MaskBit achieves state-of-the-art performance, with an FID score of 1.52 on the popular ImageNet $256 \times 256$ image generation benchmark using a compact generator model of 305M parameters. All implementations will be made publicly available to support further research in this field.

Our contribution can be summarized as follows:

1. We study the key ingredients of modern closed-source VQGAN tokenizers, and develop an open-source and high-performing VQGAN model, called VQGAN+, achieving a significant improvement of 6.28 rFID over the original VQGAN developed three years ago.

2. Building on our improved tokenizer framework, we leverage Lookup-Free Quantization (LFQ). We observe that embedding-free bit token representation exhibits highly structured semantics, important for generation.

3. Motivated by these discoveries, we develop a novel embedding-free generation framework, MaskBit, which achieves state-of-the-art performance on the ImageNet $256 \times 256$ image generation benchmark.

## 2 Revisiting VQGAN

Image generation methods (Esser et al., 2021; Rombach et al., 2022) that operate in a latent space rather than in the pixel space require networks to project images into this latent space and then back to the pixel space. Such networks are usually trained with a reconstruction objective, as is common in VAE-inspired methods (Kingma & Welling, 2014; Oord et al., 2017; Esser et al., 2021). Over time, the training frameworks for these networks have become increasingly complex (Esser et al., 2021; Yu et al., 2022a; Chang et al., 2022; Lee et al., 2022a; Yu et al., 2023; 2024a). Despite this complexity, the exploration and optimization of these frameworks remain significantly underrepresented in the scientific literature. In particular, issues in reproduction arise when crucial details are not transparently discussed, making a comparison under fair and consistent conditions hard. Moreover, fully disclosing all details enables the community to scrutinize and validate the true advancements brought by new designs. In response to this challenge, we conduct a systematic study on the essential modifications to the popular baseline Taming-VQGAN (Esser et al., 2021). For this study, we use the term "VQGAN" to refer to any network of this type and "Taming-VQGAN" specifically to refer to the network published by (Esser et al., 2021). In the following, we provide an empirical roadmap for developing a modernized VQGAN, named VQGAN+, aiming to bridge the performance gap and make advanced image generation techniques more accessible.

### 2.1 Preliminaries

Variational Autoencoders (VAEs) (Kingma & Welling, 2014) compress high-dimensional data into low-dimensional representations using an encoder network and reverse this process with a decoder network. VAEs are typically trained with a reconstruction loss, such as the $L_2$ loss. However, minimizing the $L_2$ distance alone is insufficient for achieving visually satisfying results. Therefore, a perceptual loss term (Johnson et al., 2016) based on the LPIPS metric (Zhang et al., 2018) is used to reduce noise and improve image quality. Vector-Quantized VAEs (VQ-VAEs) (Oord et al., 2017) incorporate a learnable codebook that acts as a lookup table between the encoder and decoder. The codebook is trained with two $L_2$ losses: the commitment loss, which reduces the distance of the encoder output to the codebook entries, and the codebook loss, which minimizes the distance from the codebook entries to the encoder output. VQGANs (Esser et al., 2021) build on top of VQ-VAEs by incorporating a discriminator network to add an adversarial objective to the training

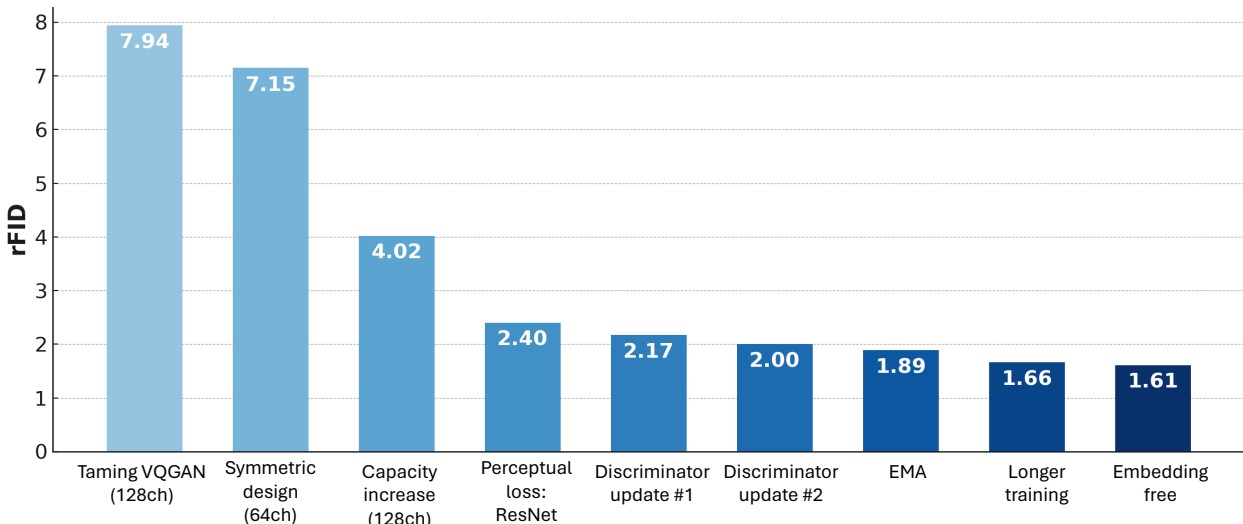

Figure 2: **Detailed roadmap to build a modern VQGAN+.** This overview summarizes the performance gains achieved by each proposed change to the architecture and training recipe. The reconstruction FID (rFID) is computed against the validation split of ImageNet at a resolution of 256. The popular and open-source Taming-VQGAN (Esser et al., 2021) serves as the baseline and starting point.

framework (Goodfellow et al., 2014). The discriminator's goal is to distinguish reconstructed images from original ones, while the VQ-VAE is trained to fool the discriminator.

## 2.2   The Experimental Setup

We follow standard practices to train and evaluate the network on ImageNet (Deng et al., 2009). Unless specified otherwise, all networks are trained with a batch size of 256 for 300,000 iterations. Evaluation is performed using the reconstruction Fréchet Inception Distance (rFID) (Heusel et al., 2017), where lower values indicate better performance. We use Taming-VQGAN (Esser et al., 2021) as our baseline and starting point, due to its thorough study and open-source availability, ensuring reproducibility of results. Taming-VQGAN is an encoder-decoder network that uses residual blocks (He et al., 2016) with convolutions for low-level feature processing and interleaved blocks of convolutions and self-attention (Vaswani et al., 2017) for high-level feature processing. In our experiments, we focus on encoder networks that produce latent embeddings with an output stride of 16, resulting in $16 \times 16$ latent embeddings for input images of $256 \times 256$ pixels. The learned vocabulary consists of 1,024 entries, each being a 256-dimensional vector.

## 2.3   VQGAN+: A Modern VQGAN

Starting with the baseline Taming-VQGAN (Esser et al., 2021) that attains 7.94 rFID on ImageNet-1K $256 \times 256$ (Deng et al., 2009), we provide a step-by-step walk-through and ablation of all changes in the following paragraphs. A summary of the changes and their effects is given in Fig. 2.

**Basic Training Recipe and Architecture.**   The initial modifications to the Taming-VQGAN baseline are as follows: (1) removing attention blocks for a purely convolutional design, (2) adding symmetry to the generator and discriminator, and (3) updating the learning rate scheduler. Removing the attention layers, as adopted in recent methods (Chang et al., 2022; Yu et al., 2023; 2024a), reduces computational complexity without sacrificing performance. We also observed discrepancies between the open-source implementation of Taming-VQGAN (Esser et al., 2021) and its description in the publication. The implementation uses two residual blocks per stage in the encoder but three per stage in the decoder. For our experiments, we adopt a symmetric architecture with two residual blocks per stage in both the encoder and decoder, as described in the publication. Additionally, we align the number of base channels in the generator and discriminator.

Specifically, we reduce the channel dimension of the generator from 128 to 64 to match the discriminator. Here, base channels refer to the feature dimension in the first stage of the network. The third modification involves the learning rate scheduler. We replace a constant learning rate with a cosine-learning rate scheduler with warmup (Loshchilov & Hutter, 2016). Specifically, we use a warmup phase of 5,000 iterations to reach the initial learning rate of $1e-4$ and then gradually decay it to 10% of the initial learning rate over time.

$\rightarrow$ *Result: 7.15 rFID (an improvement of 0.79 rFID)*

**Increasing Model Capacity.** Next, we increase the number of base channels from 64 to 128 for both generator and discriminator. This adjustment aligns the generator's capacity with that of the Taming-VQGAN baseline, except for the absence of attention blocks. As a result, the generator uses 17 million fewer parameters overall compared to the Taming-VQGAN baseline, while the number of parameters in the discriminator increases by only 8.3 million.

$\rightarrow$ *Result: 4.02 rFID (an improvement of 3.13 rFID)*

**Perceptual Loss.** The perceptual loss (Johnson et al., 2016; Zhang et al., 2018) plays a crucial role in further reducing the rFID score. In Taming-VQGAN, the LPIPS (Zhang et al., 2018) score obtained by the LPIPS VGG (Simonyan & Zisserman, 2015) network is minimized to improve image decoding. However, we noticed in the partially open-source code of recent work on image generation (Yu et al., 2023)[1], the usage of a ResNet50 (He et al., 2016) network instead to compute the perceptual loss. Although confirmed by the authors of (Yu et al., 2023; 2024a), this change has so far not been discussed in the literature. We believe this change was already introduced in earlier publications of the same group (Chang et al., 2022), but never transparently documented. Following the open-source code of (Yu et al., 2023), we apply an $L_2$ loss on the logits of a pretrained ResNet50 using the original and reconstructed images. While incorporating intermediate features into this loss can improve reconstruction further, it negatively affects generation performance in our experiments. Therefore, we compute the loss solely on the logits.

$\rightarrow$ *Result: 2.40 rFID (an improvement of 1.62 rFID)*

**Discriminator Update.** The original PatchGAN discriminator (Esser et al., 2021) employs $4 \times 4$ convolutions and batch normalization (Ioffe & Szegedy, 2015), resulting in an output resolution of $30 \times 30$ from a $256 \times 256$ input and utilizing 11 million parameters (with 128 base channels). We propose to replace the $4 \times 4$ convolution kernels with $3 \times 3$ kernels and switch to group normalization (Wu & He, 2018), producing a $16 \times 16$ output resolution to align the output stride between the generator and discriminator. These adjustments reduce the discriminator's parameter count to 7.4 million. In the second update, following prior work (Yu et al., 2024a), we replace average pooling for downsampling with a precomputed $4 \times 4$ Gaussian blur kernel using a stride of 2 (Zhang, 2019). Additionally, we incorporate LeCAM loss (Tseng et al., 2021) to stabilize adversarial training.

$\rightarrow$ *Result: 2.00 rFID (an improvement of 0.4 rFID)*

**The Final Changes.** Our last addition to the training framework is the use of Exponential Moving Average (EMA) (Polyak & Juditsky, 1992). We found that EMA significantly stabilizes the training and improves convergence, while also providing a small performance boost. Orthogonal to the reconstruction performance, we also add an entropy loss (Yu et al., 2024a) to ease the learning in the generation phase. To achieve the final performance of VQGAN+, we increase the number of training iterations from 300,000 to 1.35 million iterations, following common practices (Yu et al., 2024a; Gao et al., 2023).

$\rightarrow$ *Result: 1.66 rFID (an improvement of 0.34 rFID)*

**An Embedding-Free Variant.** Recent works (Yu et al., 2024a; Mentzer et al., 2024) have shown that removing the lookup table from the quantization Oord et al. (2017) improves scalability. Following the Lookup-Free Quantization (LFQ) approach (Yu et al., 2024a), we implement a binary quantization process by projecting the latent embeddings to $K$ dimensions ($K = 12$ in this experiment) and then quantizing them based on their sign values. Due to this binary quantization, we can directly interpret the obtained quantized

---

[1] https://github.com/google-research/magvit/

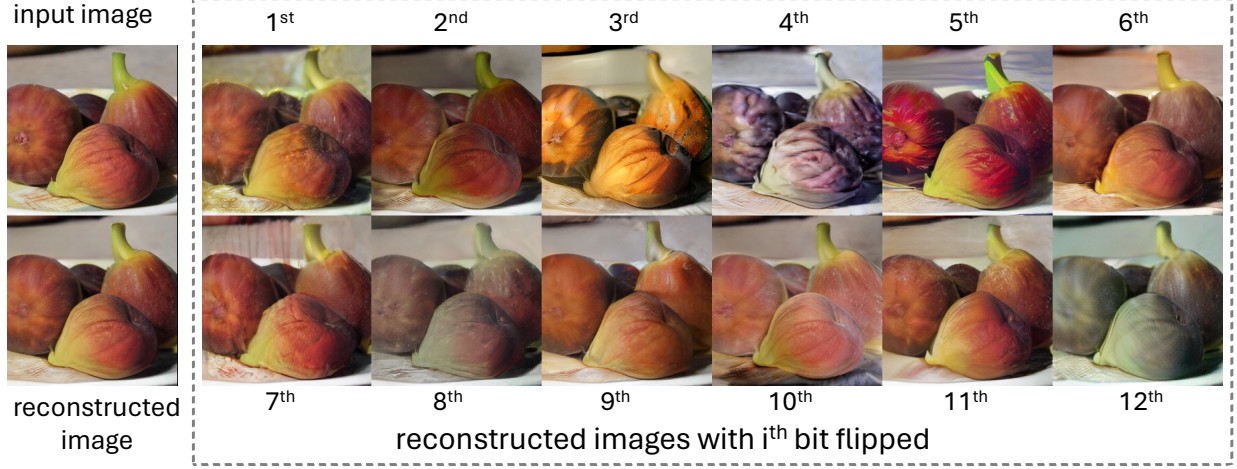

Figure 3: **Bit tokens exhibit structured semantic representations.** We visualize images obtained after corrupting tokens via bit flipping. Specifically, we utilize our method to encode images into bit tokens, where each token is represented by $K$-bits ($K = 12$ in this example). We then flip the $i$-th bit for all the bit tokens and reconstruct the images as usual. Interestingly, the reconstructed images from these bit-flipped tokens remain semantically consistent to the original image, exhibiting only minor visual modifications such as changes in texture, exposure, smoothness, color palette, or painterly quality.

representations as their token numbers. We use the standard conversion between base-2 and base-10 integers (*e.g.*, $1001_2 = 9_{10}$). In VQGAN+, the token is merely the index number of the embedding table; however, in this case, the token itself contains its corresponding representation, eliminating the need for an additional embedding. Implicitly, this defines a non-learnable codebook of size $2^K$. Given the compact nature of binary quantization with only $K$ feature dimensions, we do not anticipate significant gains in reconstruction. However, we will empirically demonstrate the clear advantages of this representation in the generation stage.

$\rightarrow$ *Result: 1.61 rFID (an improvement of 0.05 rFID)*

## 3 MaskBit: A New Embedding-free Image Generation Model

Since the typical pipeline of VQGAN-based image generation models (Chang et al., 2022; Yu et al., 2024a) includes two stages with Stage-I already discussed in the previous section, we focus on the Stage-II in this section. Stage-II refers to a transformer network that learns to generate images in the VQGAN's latent space. During Stage-II training, the tokenizer produces latent representations of images, which are then masked according to a predefined schedule (Chang et al., 2022). The masked tokens are fed into the Stage-II transformer network, which is trained with a categorical reconstruction objective for the masked tokens. This process resembles masked language modeling, making language models such as BERT (Devlin et al., 2018) suitable for this stage. These transformer models "relearn" a new embedding for each token, as the input tokens themselves are just arbitrary index numbers.

**Bit Tokens Are Semantically Structured.** Following our study of embedding-free tokenization, which eliminates the lookup table from Stage-I (*i.e.*, removing the need for embedding tables in VQ-VAE (Oord et al., 2017)) by directly quantizing the tokens into $K$-bits, we study the achieved representation more carefully. As shown in (Yu et al., 2024a), this representation yields high-fidelity image reconstruction results, indicating that images can be well represented and reconstructed using bit tokens. However, prior arts (Yu et al., 2024a) have used different token spaces for Stage-I and Stage-II, intuitively allowing the Stage-II token to focus more on semantics, while enabling the Stage-I token space to capture all visual details necessary for faithful reconstruction. This raises the question:

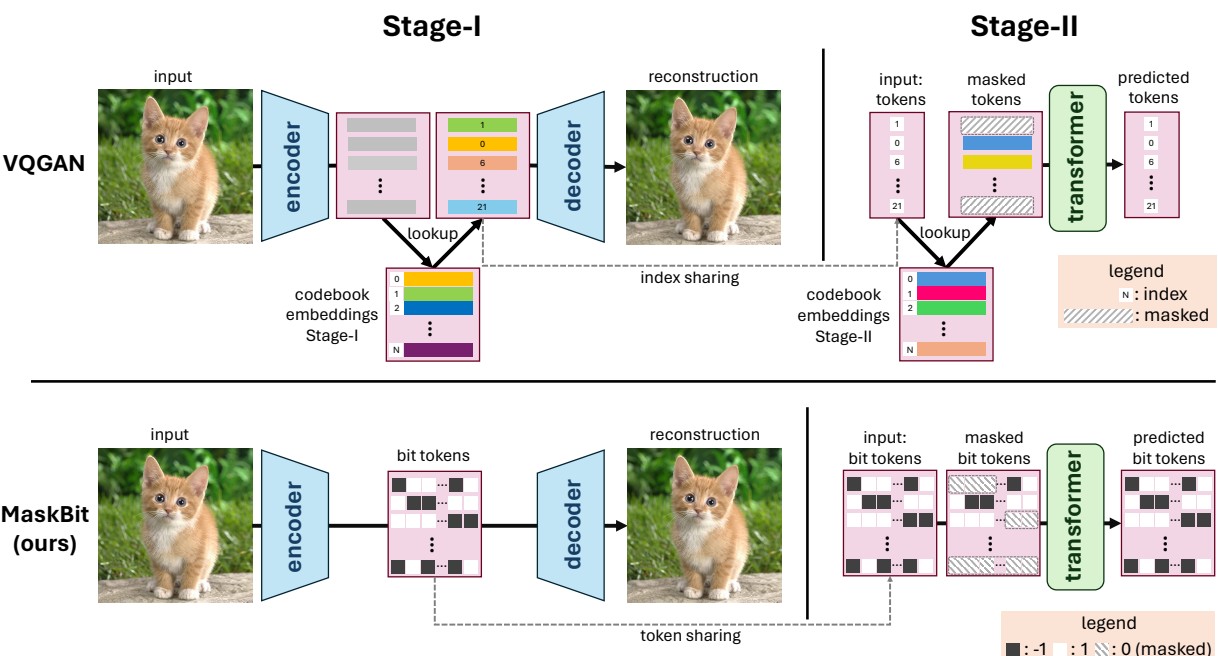

Figure 4: **High-level comparison of the architectures.** Our training framework comprises two stages for image generation. In Stage-I, an encoder-decoder network compresses images into a latent representation and decodes them back. Stage-II masks the tokens, feeds them into a transformer and predicts the masked tokens. Most prior art uses VQGAN-based methods (top) that learn independent embedding tables in both stages. In VQGAN-based methods, only indices of embedding tables are shared across stages, but not the embeddings. In MaskBit, however, neither Stage-I nor Stage-II utilizes embedding tables. The Stage-I predicts bit tokens by using binary quantization on the encoder output directly. The Stage-II partitions the shared bit tokens into groups, masks and feeds them into a transformer to predict the masked bit tokens. An illustration of grouping and masking of bit tokens can be found in the Appendix C.

*Are bit tokens also a good representation for generation, and can bit tokens eliminate the need for "re-learning" new embeddings in the typical Stage-II network?*

To address this, we first analyze the bit tokens' learned representations for image reconstruction. Specifically, given a latent representation with 256 tokens from our model (where each token is represented by $K$-bits, and each bit by either $-1$ or $1$), we conduct a a bit flipping experiment to show how the structure of the token space relates to the content. This test involves flipping the $i$-th bit (*i.e.*, swapping $-1$ and $1$) for all 256 bit tokens and decoding them as usual to produce images. The bit-flipped tokens each have a Hamming distance of 1 from the original bit tokens. Surprisingly, we find that the reconstructed images from these bit-flipped tokens are still visually and semantically similar to the original images. This indicates that the representation of bit tokens has learned structured semantics, where neighboring tokens (within a Hamming distance of 1) are semantically similar to each other. We visualize an example of this in Fig. 3 and provide more examples in Sec. E. We note that the structural differences in the latent space between bit tokens and learnable embeddings also have implications on the generator's representation, which we describe in Appendix D. Since the generation model is usually semantically conditioned on external input such as classes, the model needs to learn the semantic relationships of tokens. Given this evidence of the inherent structured semantic representation in bit tokens, we believe that the learned rich semantic structure in the bit token representation can thus be a good representation for the Stage-II generation model.

**Proposing MaskBit.** This motivates us to propose MaskBit for Stage-II, a transformer-based generative model that generates images directly using bit tokens, removing the need for the embedding lookup table as

required by existing methods (Chang et al., 2022; Yu et al., 2024a). Our training framework is illustrated in Fig. 4. Prior art using VQGAN approaches share only indices of the embedding tables between the two stages, but learn independent embeddings. Taking advantage of the built-in semantic structure of bit tokens, the proposed MaskBit can share the tokens directly between Stage-I and Stage-II. We discuss more detailed differences between the two approaches in Sec. D. Next, we provide details on bit token masking.

**Masked Bits Modeling.** The Stage-II training follows the masked modeling framework (Devlin et al., 2018), where a certain number of tokens are masked (*i.e.*, replaced with a special mask token) before being fed into the transformer, which is trained to recover the masked tokens. This approach requires an additional entry in the embedding table to learn the embedding vector for the special mask token. However, this presents a challenge for an embedding-free setup, where images are generated directly using bit tokens without embedding lookup. Specifically, it raises the question of how to represent the masked bit tokens in the new framework. To address this challenge, we propose a straightforward yet effective solution: *using zeros to represent the masked bit tokens*. In particular, a bit token $t$ is represented as $t \in \{-1, 1\}^K$ (*i.e.*, $K$-bits, with each bit being either $-1$ or $1$), while we set all masked bits to zero. Consequently, these masked bit tokens do not contribute to the image representation.

**Masked "Groups of Bits" Modeling.** During training, the network is supervised with a categorical cross-entropy loss applied to all masked tokens. With increasing number of bits, the categorical cross-entropy gets computed over an exponentially increasing distribution size. Given that bit tokens capture a channel-wise binary quantization, we explore masking "groups of bits". Specifically, for each bit token $t \in \{-1, 1\}^K$, we split it into $N$ groups $t_n \in \{-1, 1\}^{K/N}$, $\forall n \in \{1, \cdots, N\}$, with each group contains $K/N$ consecutive bits. During the masking process, each group of bits can be independently masked. Consequently, a bit token $t$ may be partially masked, allowing the model to leverage unmasked groups to predict the masked bits, easing the training process. An illustration of the grouping and masking procedure can be found in Appendix C. During the inference phase, the sampling procedure allows to sample some groups and use their values to guide the remaining samplings. However, this approach increases the number of bit token groups to be sampled, posing a challenge during inference due to the potential for poorly chosen samples. Empirically, we found that using two groups yields the best performance, striking a good balance.

## 4 Experimental Results for Image Generation

**Experimental Setup.** We evaluate the proposed MaskBit on class-conditional image generation. Specifically, the network generates a total of 50,000 samples for the 1,000 ImageNet (Deng et al., 2009) classes. All Stage-II generative networks are trained and evaluated on ImageNet at resolutions of $256 \times 256$. We report the main metric Fréchet Inception Distance for generation (gFID) (Heusel et al., 2017), along with the side metric Inception Score (IS) (Salimans et al., 2016). All scores are obtained using the official evaluation suite and reference statistics by ADM (Dhariwal & Nichol, 2021). We train all Stage-II networks for 1,080 epochs and report results with classifier-free guidance (Ho & Salimans, 2022). More training and inference details can be found in Sec. B.

### 4.1 Main Results

Tab. 1 summarizes the generation results on ImageNet $256 \times 256$. We make the following observations.

**VQGAN+ as a New Strong Baseline for Image Tokenizers.** By carefully revisiting the VQGAN design in Sec. 2, we develop VQGAN+, a solid new baseline for image tokenizers that achieves an rFID of 1.66, marking a significant improvement of 6.28 rFID over the original Taming-VQGAN model (Esser et al., 2021). Using VQGAN+ in the generative framework of MaskGIT (Chang et al., 2022) results in a gFID of 2.12, outperforming several recent advanced VQGAN-based methods, such as ViT-VQGAN (Yu et al., 2022a) (by 0.92 gFID), RQ-Transformer (Lee et al., 2022a) (by 1.68 gFID), and even the original MaskGIT (Chang et al., 2022) (by 1.90 gFID). This result shows that the improvements we propose to the Stage-I model, indeed transfer to the generation model. Furthermore, our result surpasses several recent diffusion-based generative models that utilize the heavily optimized VAE (Kingma & Welling, 2014) by Stable Diffusion (AI,

Table 1: **ImageNet** $256 \times 256$ **generation results.** †: Tokenizers trained on OpenImages (Kuznetsova et al., 2020). ‡: Tokenizers trained on OpenImages, LAION-Aesthetics/-Humans (Schuhmann et al., 2022). *: Concurrent methods. All results use classifier-free guidance or rejection sampling, but not both. #Params: Generator's parameters. #Steps: Sampling steps.

| Model | Tokenizer | | Generator | | | |
| | #tokens | codebook | gFID↓ | IS↑ | #Params | #Steps |
|---|---|---|---|---|---|---|
| *continuous latent representation* | | | | | | |
| LDM-4-G (Rombach et al., 2022)† | 4096×3 | - | 3.60 | 247.7 | 400M | 250 |
| GIVT-L-A (Tschannen et al., 2024) | 256×32 | - | 2.59 | - | 1.67B | 256 |
| U-ViT (Bao et al., 2023)‡ | 1024×4 | - | 2.29 | 263.9 | 501M | 50 |
| DiT-XL/2-G (Peebles & Xie, 2023)‡ | 1024×4 | - | 2.27 | 278.2 | 675M | 250 |
| MDT (Gao et al., 2023)‡ | 1024×4 | - | 1.79 | 283.0 | 676M | 250 |
| DiMR (Liu et al., 2024)‡* | 1024×4 | - | 1.70 | 289.0 | 505M | 250 |
| MDTv2 (Gao et al., 2024)‡* | 1024×4 | - | 1.58 | 314.7 | 676M | 250 |
| MAR-H (Li et al., 2024)* | 256×16 | - | 1.55 | 303.7 | 943M | 256 |
| *discrete latent representation* | | | | | | |
| Taming-VQGAN (Esser et al., 2021) | 256 | 1024 | 5.88 | 304.8 | 1.4B | 256 |
| MaskGIT (Chang et al., 2022) | 256 | 1024 | 4.02 | 355.6 | 177M | 8 |
| RQ-Transformer (Lee et al., 2022a) | 256 | 16384 | 3.80 | 323.7 | 3.8B | 64 |
| ViT-VQGAN (Yu et al., 2022a) | 1024 | 8192 | 3.04 | 227.4 | 1.7B | 1024 |
| LlamaGen-3B (Sun et al., 2024)* | 576 | 16384 | 2.18 | 263.3 | 3.1B | 576 |
| TiTok-S-128 (Yu et al., 2024c)* | 128 | 4096 | 1.97 | 281.8 | 287M | 64 |
| VAR (Tian et al., 2024)†* | 680 | 4096 | 1.92 | 350.2 | 2.0B | 10 |
| MAGVIT-v2 (Yu et al., 2024a) | 256 | 262144 | 1.78 | 319.4 | 307M | 64 |
| MaskGIT (w/ our VQGAN+) | 256 | 4096 | 2.12 | 300.8 | 310M | 64 |
| MaskBit (ours) | 256 | 4096 | 1.65 | 341.8 | 305M | 64 |
| MaskBit (ours) | 256 | 16384 | 1.62 | 338.7 | 305M | 64 |
| MaskBit (ours) | 256 | 16384 | 1.52 | 328.6 | 305M | 256 |

2022), including LDM (Rombach et al., 2022), U-ViT (Bao et al., 2023), and DiT (Peebles & Xie, 2023). We believe these results with VQGAN+ establish a new solid baseline for the community.

**Bit Tokens as a Strong Representation for Both Reconstruction and Generation.** As discussed in previous sections, bit tokens encapsulate a structured semantic representation that retains essential information for image generation, forming the foundation of MaskBit. Consequently, MaskBit delivers superior performance, achieving a gFID of 1.65 with 12 bits and 1.62 with 14 bits, surpassing prior works. Notably, MaskBit outperforms the recent state-of-the-art MAGVIT-v2 (Yu et al., 2024a) while utilizing an implicit codebook that is $16\times$ to $64\times$ smaller. When the number of sampling steps is increased from 64 to 256—matching the steps used by autoregressive models—MaskBit sets a new state-of-the-art with a gFID of 1.52, outperforming concurrent works (Gao et al., 2024; Li et al., 2024; Liu et al., 2024; Sun et al., 2024; Tian et al., 2024; Yu et al., 2024c), while using a compact generator model of just 305M parameters. Remarkably, MaskBit, which operates with discrete tokens, surpasses methods relying on continuous tokens, such as MDTv2 (Gao et al., 2024) and MAR-H (Li et al., 2024).

**Comparison under Similar Generator Parameters.** In Tab. 2, we present a performance comparison across models of similar sizes. Previous works have demonstrated that increasing model size generally leads to superior performance (Li et al., 2024; Sun et al., 2024; Tian et al., 2024). Therefore, to provide a more comprehensive comparison, we only compare models using a similar number of parameters, but vary codebook size and sampling steps. As shown in the table, MaskBit not only achieves state-of-the-art results overall but also outperforms all other models under similar conditions. It is worth noting that all models in the comparison use a larger Stage-I (reconstruction) tokenizer than ours.

Table 2: **Comparison using similar generator model capacity ($\sim$ 300M) on ImageNet** $256 \times 256$. †: Tokenizers trained on OpenImages (Kuznetsova et al., 2020). *: Concurrent methods. All results use classifier-free guidance. #Steps: Sampling steps.

| | Tokenizer | | | Generator | | | |
|---|---|---|---|---|---|---|---|
| Model | #Params | #tokens | codebook | gFID↓ | IS↑ | #Params | #Steps |
| VAR (Tian et al., 2024)†* | 109M | 680 | 4096 | 3.30 | 274.4 | 310M | 10 |
| TiTok-S-128 (Yu et al., 2024c)* | 72M | 128 | 4096 | 1.97 | 281.8 | 287M | 64 |
| MaskBit (ours) | 54M | 256 | 4096 | 1.65 | 341.8 | 305M | 64 |
| LlamaGen-L (Sun et al., 2024)* | 72M | 256 | 16384 | 3.80 | 263.3 | 343M | 256 |
| MaskBit (ours) | 54M | 256 | 16384 | 1.52 | 328.6 | 305M | 256 |
| MAGVIT-v2 (Yu et al., 2024a) | 116M | 256 | 262144 | 1.78 | 319.4 | 307M | 64 |
| MaskBit (ours) | 54M | 256 | 262144 | 1.67 | 331.6 | 305M | 64 |

Table 3: **Ablation studies on ImageNet** $256 \times 256$ **generation**. Our final setting is labeled in gray. To save computations, a shorter training schedule has been used for this study.

(a) Embedding-free Schemes

| Tokenizer embedding-free | Generator embedding-free | gFID↓ |
|---|---|---|
| ✗ | ✗ | 2.12 |
| ✓ | ✗ | 1.95 |
| ✓ | ✓ | 1.82 |
| ✗ | ✓ | 2.83 |

(b) Number of Groups

| Groups | gFID↓ |
|---|---|
| 1 | 1.94 |
| 2 | 1.82 |
| 4 | 2.01 |

**Qualitative Results.** We present visualizations of the generated $256 \times 256$ images using the proposed MaskBit in Fig. 5. The results show the diversity of classes represented in ImageNet. MaskBit is able to generate high-fidelity outputs. We note that these generated results inherit the dataset bias of ImageNet. As a classification dataset, the images are centered around objects showing the respective class. Thus, this dataset bias influences the kind of images that can be generated. Methods used in production environments such as Stable Diffusion (AI, 2022) therefore train on more general large-scale datasets such as LAION-5B (Schuhmann et al., 2022) to remove such dataset bias.

## 4.2 Ablation Study

Unless stated otherwise, we use fewer training iterations for the ablation studies due to limited resources and computational costs.

**Embedding-free Stages Bring Improvements.** In Tab. 3a, we investigate embedding-free designs in both Stage-I (tokenizer) and Stage-II (generator). To ensure a fair comparison, all models use the same number of tokens, codebook size, and model capacity. In the first row, we employ VQGAN+ with a traditional Stage-II model, as in MaskGIT (Chang et al., 2022), achieving a gFID of 2.12. Using bit tokens in Stage-I improves performance to a gFID of 1.95, highlighting the effectiveness of structured semantic representation of bit tokens. Further enabling our Stage-II model MaskBit (*i.e.*, both stages now are embedding-free) achieves the best gFID of 1.82, demonstrating the advantages of embedding-free image generation from bit tokens. For completeness, we also explore VQGAN+ with an embedding-free Stage-II model, yielding a gFID of 2.83. This variant performs worse due to the lack of the semantic structure in the tokens.

**Two Groups of Bits Lead to Better Performance.** Tab. 3b presents the performance of MaskBit using a different numbers of groups. As shown in the table, when using two groups, each consisting of six consecutive bits, MaskBit achieves the best performance. This configuration outperforms a single group

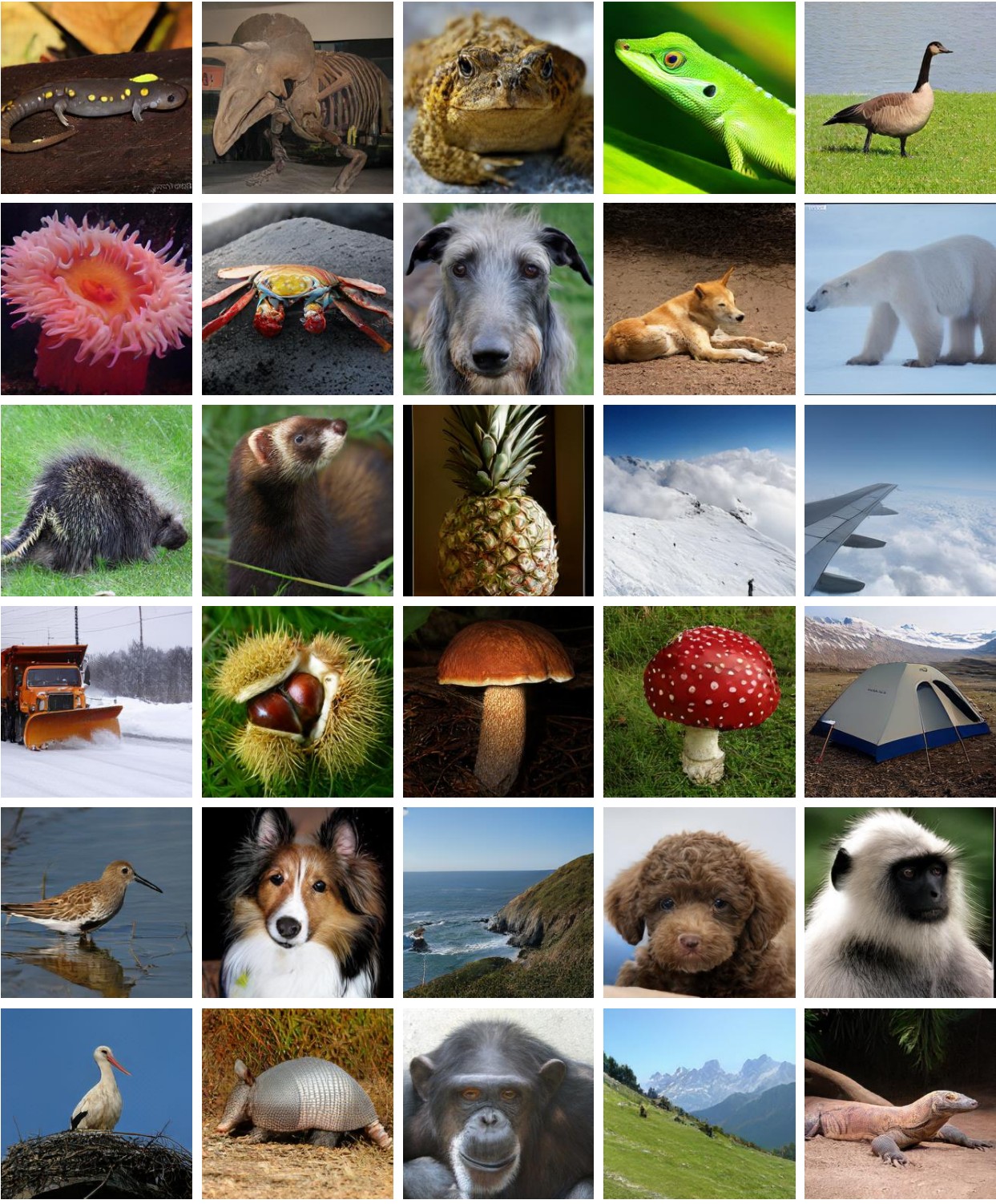

Figure 5: **Visualization of generated** $256 \times 256$ **images.** MaskBit demonstrates the ability to produce high-fidelity images across a diverse range of classes.

Table 4: **Ablation on the effect of different number of bits and different number of sampling steps on ImageNet** $256 \times 256$. Table (a) uses 64 sampling steps, and Table (b) uses a 14 bit model.

(a) Number of Bits

| Bits | #tokens | codebook | gFID↓ | IS↑ |
|------|---------|----------|-------|-----|
| 10 | 256 | 1024 | 1.68 | 353.1 |
| 12 | 256 | 4096 | 1.65 | 341.8 |
| 14 | 256 | 16384 | **1.62** | 330.7 |
| 16 | 256 | 65536 | 1.64 | 339.6 |
| 18 | 256 | 262144 | 1.67 | 331.6 |

(b) Number of Sampling Steps

| Sampling Steps | gFID↓ | IS↑ |
|----------------|-------|-----|
| 64 | 1.62 | 330.7 |
| 128 | 1.56 | 341.9 |
| 256 | 1.52 | 328.6 |

because it allows the model to leverage information from the unmasked groups. However, when using four groups, the increased difficulty in sampling during inference – due to the need for four times more sampling – detracts from performance. Still, all group configurations outperform our strong baseline using VQGAN+ as well as modern approaches like ViT-VQGAN (Yu et al., 2022a) and DiT-XL (Peebles & Xie, 2023), showing the potential of grouping. We also conduct another experiment, where we apply 2 groups to VQGAN+. The performance drops form 2.12 to 4.4 gFID, due to the missing semantic structure in the tokens.

**MaskBit with 14 Bits Yields Better Performance.** In Tab. 4a, we train the Stage-II models with our final training schedule. In each setting, MaskBit leverages a different number of bits. Empirically, we find that using 14 bits works best on ImageNet. We note that all settings outperform prior methods (*e.g.*, Peebles & Xie (2023); Yu et al. (2024a)) in gFID and the performance differences are marginal across different number of bits. While using 14 bits is sufficient for ImageNet, more general and large-scale datasets such as LAION-5B (Schuhmann et al., 2022) might benefit from using more bits (*i.e.*, larger implicit vocabulary sizes).

**MaskBit Scales with More Sampling Steps.** MaskBit follows the non-autoregressive sampling paradigm (Chang et al., 2022; Yu et al., 2023), enabling flexibility in the number of sampling steps during inference (up to 256 steps in our ImageNet 256×256 experiments). Unlike autoregressive models (Esser et al., 2021; Sun et al., 2024), this approach allows for fewer forward passes through the Stage-II generative model, reducing computational cost and inference time. However, increasing MaskBit's sampling steps to match those of autoregressive models can also improve performance. As shown in Tab. 4b, even with 64 steps, MaskBit surpasses prior work and achieves an excellent balance between compute efficiency and performance. Scaling further to 256 steps sets a new state-of-the-art gFID of 1.52.

## 5 Related Work

**Image Tokenization.** Images represented by raw pixels are highly redundant and commonly compressed to a latent space with autoencoders (Hinton & Salakhutdinov, 2006; Vincent et al., 2008), since the early days of deep learning. Nowadays, image tokenization still plays a crucial role in mapping pixels into discrete (Oord et al., 2017) or continuous (Kingma & Welling, 2014; Higgins et al., 2017) representation suitable for generative modeling. We focus on the stream of vector quantization (VQ) tokenizers, which proves successful in state-of-the-art transformer generation frameworks (Esser et al., 2021; Chang et al., 2022; Yu et al., 2022b; 2024a; Lezama et al., 2022; 2023; Lee et al., 2022a;b). Starting from the seminal work VQ-VAE (Oord et al., 2017), several key advancements have been proposed, such as the perceptual loss (Zhang et al., 2018) and discriminator loss (Karras et al., 2019) from (Esser et al., 2021) or better quantization methods (Zheng et al., 2022; Lee et al., 2022a; Zheng & Vedaldi, 2023; Mentzer et al., 2024; Yu et al., 2024a), which significantly improve the reconstruction quality. Nonetheless, modern transformer generation models (Chang et al., 2022; Yu et al., 2023; 2024a) utilize an even stronger VQ model, with details not fully discussed in the literature, resulting in a performance gap to the wider research community. In this work, we conducted a systematic study to modernize the VQGAN model, aiming at demystifying the process to obtain a strong tokenizer.

**Image Generation.** Although various types of image generation models exist, mainstream state-of-the-art methods are usually either diffusion-based (Dhariwal & Nichol, 2021; Chen et al., 2023; Rombach et al., 2022; Peebles & Xie, 2023; Li et al., 2023; Hoogeboom et al., 2023), auto-regressive-based (Chen et al., 2020; Esser et al., 2021; Yu et al., 2022a; Gafni et al., 2022; Tschannen et al., 2024; Yu et al., 2024b), or masked-transformer-based (Chang et al., 2022; Lee et al., 2022b; Yu et al., 2024a; Lezama et al., 2022; 2023). Among them, the masked-transformer-based methods exhibit competitive performance, while bringing a substantial speed-up, thanks to a much fewer sampling steps. This research line begins with MaskGIT (Chang et al., 2022), which generates images from a masked sequence in non-autoregressive manner where at each step multiple token can be predicted (Devlin et al., 2018). Follow-up work has successfully extend the idea to the video generation domain (Yu et al., 2023; Gupta et al., 2023), large-scale text-to-image generation (Chang et al., 2023), along with competitive performance to diffusion models (Yu et al., 2024a) for image generation. Still, these works only utilize the tokens from the tokenizer to learn new embeddings in the transformer network, while we aim at an embedding-free generator framework, making better use of the learned semantic structure from the tokenizer.

## 6 Conclusion

We conducted systematic experiments and provided a thorough, step-by-step study towards a modernized VQGAN+. Together with bit tokens, we provide two strong tokenizers with fully reproducible training recipes. Furthermore, we propose an embedding-free generation model MaskBit, which makes full use of the rich semantic information from bit tokens. Our method demonstrates state-of-the-art results on the challenging $256 \times 256$ class-conditional ImageNet benchmark.

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

## Appendix

In the appendix, we provide additional information as listed below:

- Sec. A provides the dataset information and licenses.

- Sec. B provides all implementation details.

- Sec. C provides a visual explanation on masking groups of bit tokens.

- Sec. D provides more discussion on bit tokens.

- Sec. E provides more reconstructed images for the bit flipping experiments on ImageNet and COCO.

- Sec. F provides zero-shot reconstruction results on COCO.

- Sec. G provides a nearest-neighbor analysis on ImagetNet.

- Sec. H discusses the method's limitations.

- Sec. I discusses the broader impacts.

## A   Datasets

**ImageNet:**   ImageNet (Deng et al., 2009) is one of the most popular benchmarks in computer vision. It has been used to benchmark image classification, class-conditional image generation, and more.

License: Custom License, non-commercial. `https://image-net.org/accessagreement`

Dataset website: `https://image-net.org/`

## B   Implementation Details

In the following, we provide implementation details for the purpose of reproducibility:

### B.1   Stage-I

For data augmentation, we closely follow the prior arts (Yu et al., 2023; 2024a). Specifically, we use random cropping of 80%-100% of the original image with a random aspect ratio of 0.75-1.33. After that, we resize the input to $256 \times 256$ and apply horizontal flipping.

- Base channels: 128

- Base channel multiplier per stage: [1,1,2,2,4]

- Residual blocks per stage: 2

- Adversarial loss enabled at iteration: 20000

- Discriminator loss weight: 0.02

- Discriminator loss: hinge loss

- Perceptual loss weight: 0.1

- LeCam regularization weight: 0.001

- L2 reconstruction weight: 4.0

- Commitment loss weight: 0.25

- Entropy loss weight: 0.02

- Entropy loss temperature: 0.01

- Gradient clipping by norm: 1.0

- Optimizer: AdamW (Loshchilov & Hutter, 2019)

- Beta1: 0.9

- Beta2: 0.999

- Weight decay: 1e-4

- LR scheduler: cosine

- Base LR: 1e-4

- LR warmup iterations: 5000

- Training iterations: 1350000

- Total Batchsize: 256

- GPU: A100

### B.2   Stage-II

For data augmentation, we closely follow the prior arts (Yu et al., 2023; 2024a). Specifically, we use random cropping of 80%-100% of the original image. After that, we resize the input to $256 \times 256$ and apply horizontal flipping.

- Hidden dimension: 1024

- Depth: 24

- Attention heads: 16

- MLP dimension: 4096

- Dropout: 0.1

- Mask schedule: arccos

- Class label dropout: 0.1

- Label smoothing: 0.1

- Gradient clipping by norm: 1.0

- Optimizer: AdamW (Loshchilov & Hutter, 2019)

- Beta1: 0.9

- Beta2: 0.96

- Weight decay: 0.045

- LR scheduler: cosine

- Base LR: 1e-4

- LR warmup iterations: 5000

- Training iterations: 1350000

- Total Batchsize: 1024

- GPU: A100

We use 32 A100 GPUs for training Stage-I models. Training time varies with respect to the number of training iterations between 1 (300K iterations) and 4.5 (1.35M iterations) days. We note that the Stage-I model can also be trained with fewer GPUs without any issues, but this will accordingly increase the training duration. Stage-II models are trained with 64 A100 GPUs and take 4.2 days for the longest schedule (1.35M iterations).

## B.3 Sampling Procedure

Our sampling procedure closely follows prior works for non-autoregressive image generation (Gao et al., 2023; Chang et al., 2022; Yu et al., 2024a; 2023; Chang et al., 2023). By default, the Stage-II network generates images in 64 steps. Each step, a number of tokens is unmasked and kept. The number of tokens to keep is determined by an arccos scheduler (Besnier & Chen, 2023; Chang et al., 2022). Additionally, we use classifier-free guidance and a guidance scheduler (Gao et al., 2023; Yu et al., 2023). We refer to the codebase available on `https://github.com/markweberdev/maskbit/` for the detailed sampling parameters for each model.

## B.4 Sensitivity to Sampling Parameters

In this subsection, we detail the specific values for the sampling hyper-parameters to ensure reproducibility. However, the reported FID scores can be obtained by a large range of hyper-parameters. Due to the inherent randomness in the sampling procedure, results might vary by $\pm 0.02$ FID. For the resolution of $256 \times 256$, we verified several choices of parameters obtaining similar results: temperature $t \in [7.3, 8.3]$, guidance $s \in [5.8, 7.8]$ and "scale_pow" $m \in [2.5, 3.5]$.

# C Visualization of Masking Groups of Bit Tokens

Fig. 6 provides a visualization of the approach, masking groups of bit tokens, presented in this paper. When using a single group, a bit token will either be masked completely or remain unchanged after the masking is applied. When using more than a single group, the bit token is split into groups of consecutive bits. Each of these groups can be independently masked during the masking procedure, leading to some cases where the bit tokens are partially masked as shown in the illustration. Notably, within each group of bits, the masking can only be applied to all bits in the group.

# D More Discussion of Bit Tokens

## D.1 Connection of Semantic Structure and Generator Performance

Prior work (Gu et al., 2024) studying the tokenizer objective has found that a strong semantic representation in the latent space is important for good generation performance. This observation was made by using different variations of the LPIPS perceptual loss. The experiments presented in this work in Sec. 2 support the same observation. Specifically, VQGAN+ uses a ResNet logits-based perceptual loss which we found to work well for generation. However, as mentioned in the paragraph "Perceptual Loss", adding intermediate activations of ResNet to the perceptual loss decreased the generation performance even though it helps the reconstruction performance.

Given the insights on the importance of semantics, this work studies the latent space of bit tokens and the connection to the image content. The key finding is that bit tokens exhibit this uniquely constrained structure by design, which leads to a *structured semantic representation*. In the next paragraph, we discuss how this structured representation affects the internal generation model feature representation per token.

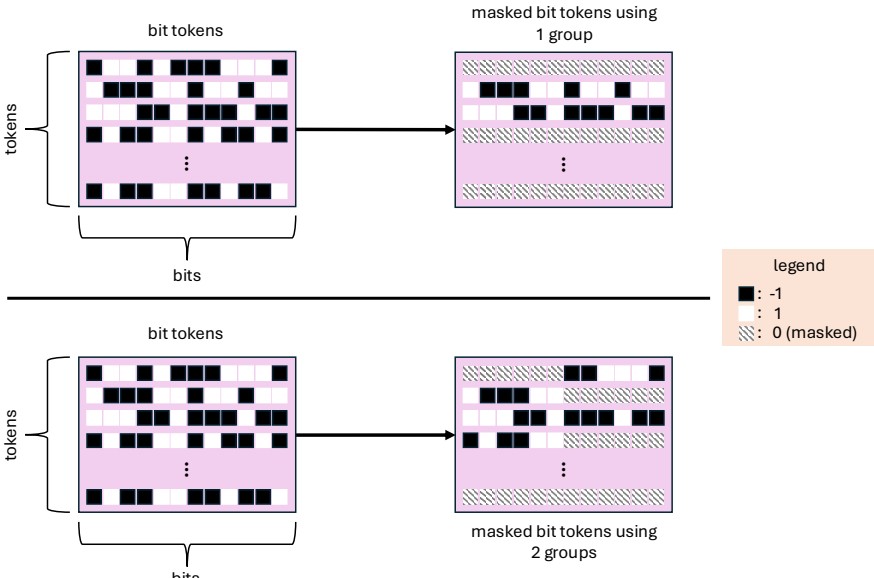

Figure 6: **Visualization of the process for masking of groups of bit tokens**. The figure visualizes the masking procedure of bit tokens (12 bits) when using either one group (*top* row) or two groups (*bottom* row). When using a single group, a bit token can either be fully masked or fully non-masked. When using two groups, each bit token is split into two consecutive groups (*i.e.*, the first group contains the first 6 bits and the second group has the last 6 bits). These two groups can be independently masked. This leads to the case where a bit token can be partially masked, as shown in the illustration.

We refer to the experimental results presented in Sec. 4.1 for a quantitative study on how much benefit bit tokens with this structured semantic representation bring to the generation performance.

## D.2 Implications of Different Token Spaces on the Generator

In prior work, only the token indices were shared between Stage-I and Stage-II models. In all cases, the Stage-II model relearns an embedding table. Considering our setting of 12 bits, a MaskGIT Stage-II model would learn a new embedding table with $2^{12} = 4096$ entries. Each of these entries is an (arbitrary) vector in $\mathbb{R}^{1024}$ with 1024 being the hidden dimension. We note that this approach is equivalent to replacing the embedding table with a linear layer processing one-hot encoded token indices. The linear layer would in this case do the same mapping as the embedding table and have the same number of parameters, *i.e.*, $4096 \times 1024$, omitting the bias for simplicity.

In MaskBit, instead of token indices, the bit tokens are shared and processed directly. MaskBit also uses a linear layer to map to the model's internal representation, but this mapping is from 12 dimensions (when $K = 12$) to 1024, which corresponds to learning just 12 embedding vectors. Thus, the learnable mapping is *reduced by a factor of 341*. This is possible as MaskBit does not use a one-hot encoded representation, but a bit token representation with 12 bits. The linear layer therefore adds/subtracts the learnable weights depending on the bit coefficients $\pm 1$. It follows that *the internal representation per token in MaskBit is not arbitrary in $\mathbb{R}^{1024}$, but significantly constrained*. These constraints require a semantically meaningful structure in the bit tokens, while in MaskGIT the representation for each token index is independent and arbitrarily learnable.

Example: Given two bit tokens $t_1$ and $t_2$ with only a difference in the last bit,

Table 5: **Zero-shot reconstruction evaluation on COCO with** $256 \times 256$ **resolution**.

| Model | zero-shot rFID $\downarrow$ |
|---|---|
| Taming-VQGAN (Esser et al., 2021) | 16.3 |
| VQGAN+ | 8.7 |
| Embedding-free variant | 8.3 |

$$t_1 = \begin{pmatrix} +1 \\ \vdots \\ -1 \\ +1 \\ +1 \\ -1 \end{pmatrix} \qquad t_2 = \begin{pmatrix} +1 \\ \vdots \\ -1 \\ +1 \\ +1 \\ +1 \end{pmatrix}$$

In MaskBit, the only difference when processing the tokens in the first linear layer is that the last weight vector, corresponding to the last bit in the tokens, is either added (for $t_2$) or subtracted (for $t_1$). Thus, the obtained results are not independent of each other and not arbitrary, which is different from prior work.

## E    More Reconstruction Results of Flipping Bit Tokens

We present additional reconstruction results from the bit flipping analysis. As shown in Fig. 7, the reconstructed images from these bit-flipped tokens remain visually and semantically similar to the original images, consistent with the findings in the main paper.

Furthermore, we repeat this experiment in a zero-shot manner on COCO (Lin et al., 2014) with the same model only trained on ImageNet. The results are visualized in Fig. 8, demonstrating that the bit token's structured semantic representation also holds for COCO images.

## F    Zero-Shot Reconstruction Results on COCO

In Tab. 5, we provide *zero-shot* evaluation of the presented tokenizers of Sec. 2 on COCO (Lin et al., 2014) to show how the modifications made to the orginal VQGAN (Esser et al., 2021) impact the performance on more complex datasets. For this experiment, we use the same models as for the experiments in Sec. 2 that were only trained on ImageNet (Deng et al., 2009). All images are resized with their shorter edge to 256 and then center cropped before feeding them into the models. The results show that the modifications made to obtain VQGAN+ and its embedding-free variant also achieve improvements on more complex datasets in the zero-shot evaluation setting.

## G    Nearest Neighbor Analysis of Generated Images

Following the idea presented in Esser et al. (2021), we conduct a nearest neighbor analysis. In this analysis, a generated image is used and compared to all images in the training set. For this comparison, we use two distance measures: LPIPS and Hamming distance. The LPIPS metric is used to measure perceptual similarity of two images, while the Hamming distance takes the bit tokens of the encoded images and computes the bit-wise distance. In Fig. 9 and Fig. 10, we visualize the 10 nearest neighbors for each generated image and distance measure. The results show that the model is not just memorizing samples from the training set, but generates indeed new images. Since LPIPS and Hamming distance are two different measures, the obtained nearest neighbors also differ. For example, considering the neighbors to the generated images of the parrot or the space rocket in Fig. 9, we observe that the nearest samples according to LPIPS focus more on staying true to the reference color palette, while this is less the case for the closest samples according to the Hamming distance. This finding aligns with the observations in the bit-flipping experiment,

where with small changes in the Hamming distance, the visual attributes can vary quite a bit, while the semantic content stays the same.

## H  Limitations

In this study, the focus has been on non-autoregressive, token-based transformer models for generation. Due to the computational costs, we did not explore using the modernized VQGAN model (VQGAN+) in conjunction with diffusion models. For similar reasons, MaskBit was only trained on ImageNet, thus inheriting the dataset bias, *i.e.*, focus on a single foreground object. This makes it difficult to do have a qualitative comparison to methods trained on large-scale datasets such as LAION (Schuhmann et al., 2022). We also note, that the standard benchmark metric FID has received criticism due to it measuring only distribution similarities. However, due to the lack of better and established alternatives, this study uses FID for its quantitative evaluation and comparison.

## I  Broader Impacts

Generative image models have diverse applications with significant social impacts. Positively, they enhance creativity in digital art, entertainment, and design, enabling easier and potentially new forms of expression, art and innovation. However, these models pose risks, including the potential for misuse such as deepfakes, which can spread misinformation and facilitate harassment. Additionally, generative models can suffer from social and cultural biases present in their training datasets, reinforcing negative stereotypes. We notice that these risks have become a dilemma for open-source and reproducible research. The scope of this work however is purely limited to class-conditional image generation. In class-conditional generation, the model will only be able to generate images for a predefined, public and controlled set of classes. In this case, the images and class set are coming from ImageNet. As a result, we believe that this specific work does not pose the aforementioned risks and is therefore safe to share.

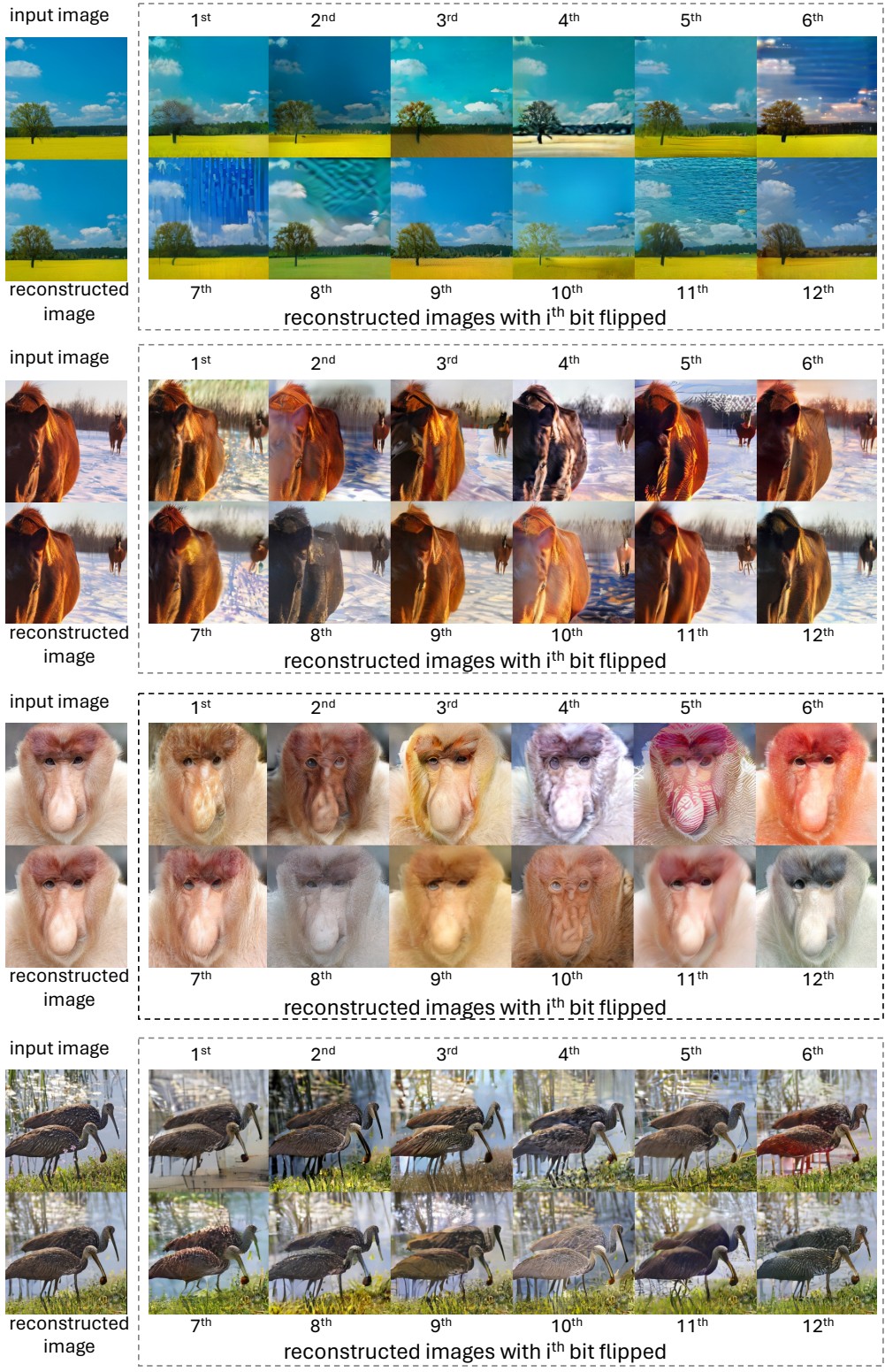

Figure 7: **More examples demonstrating the structured semantic representation in bit tokens.** Similar to the observation in the main paper, the reconstructed images from these bit-flipped tokens remain semantically similar to the original image, exhibiting only minor visual modifications such as changes in texture, exposure, smoothness, color palette, or painterly quality.

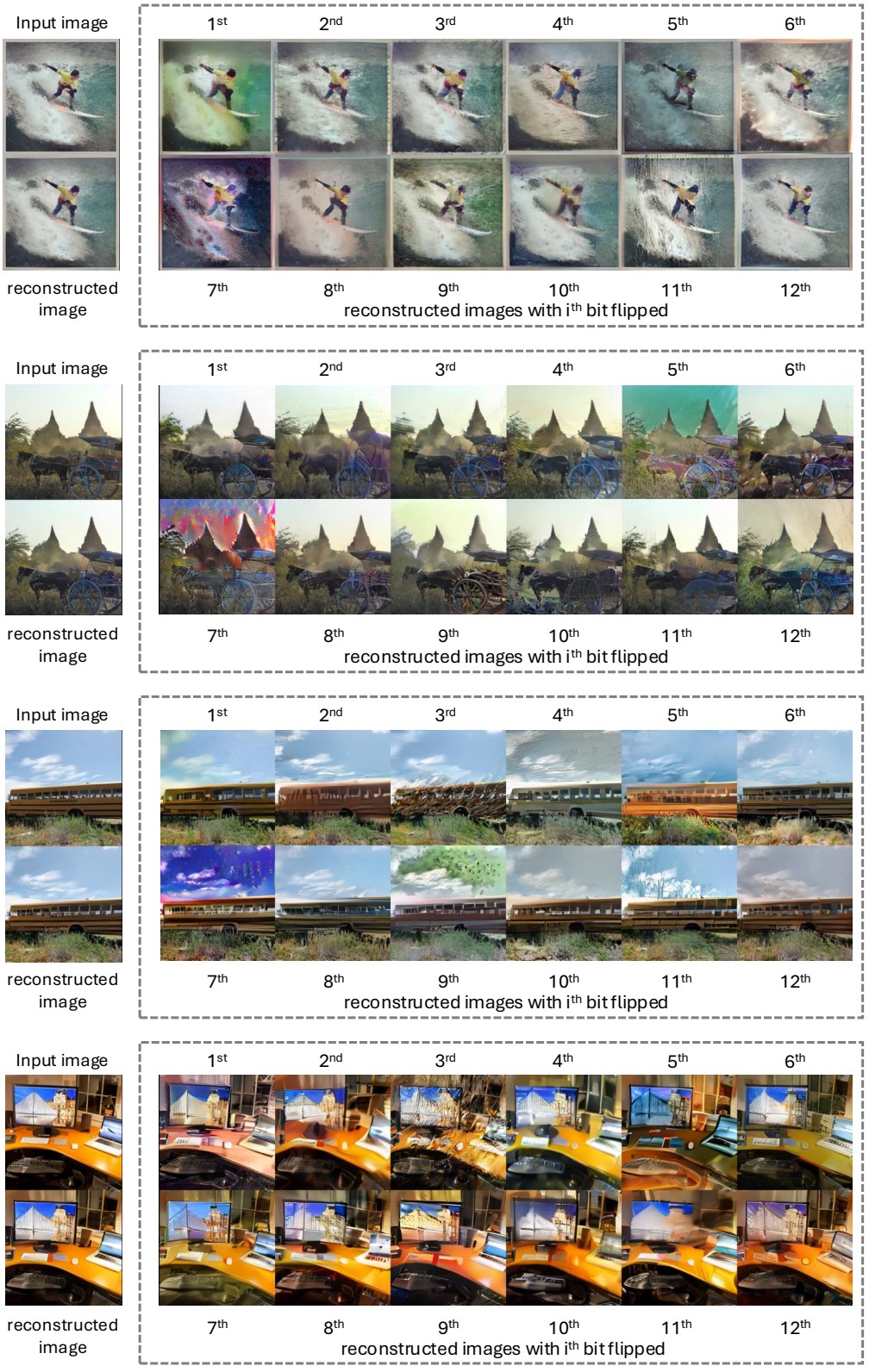

Figure 8: **Examples demonstrating the structured representation of bit tokens in a zero-shot setting.** The structured semantic representation can also be found when running the tokenizer in a zero-shot manner on the more complex dataset COCO (Lin et al., 2014).

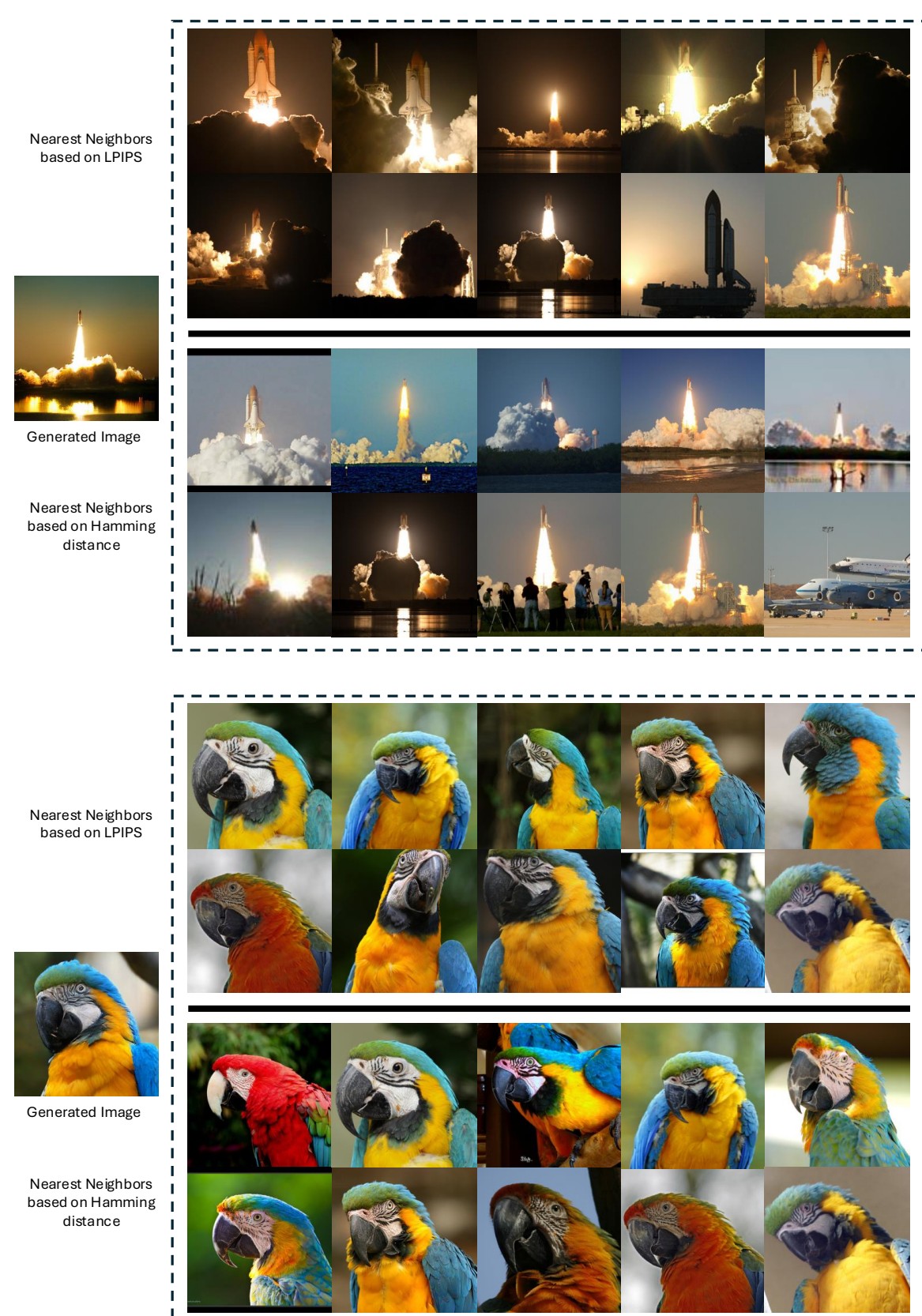

Figure 9: **Nearest Neighbors of two generated Images.** The figure shows the ten nearest neighbors in the ImageNet training set in terms of LPIPS (*top*) and Hamming distance (*bottom*).

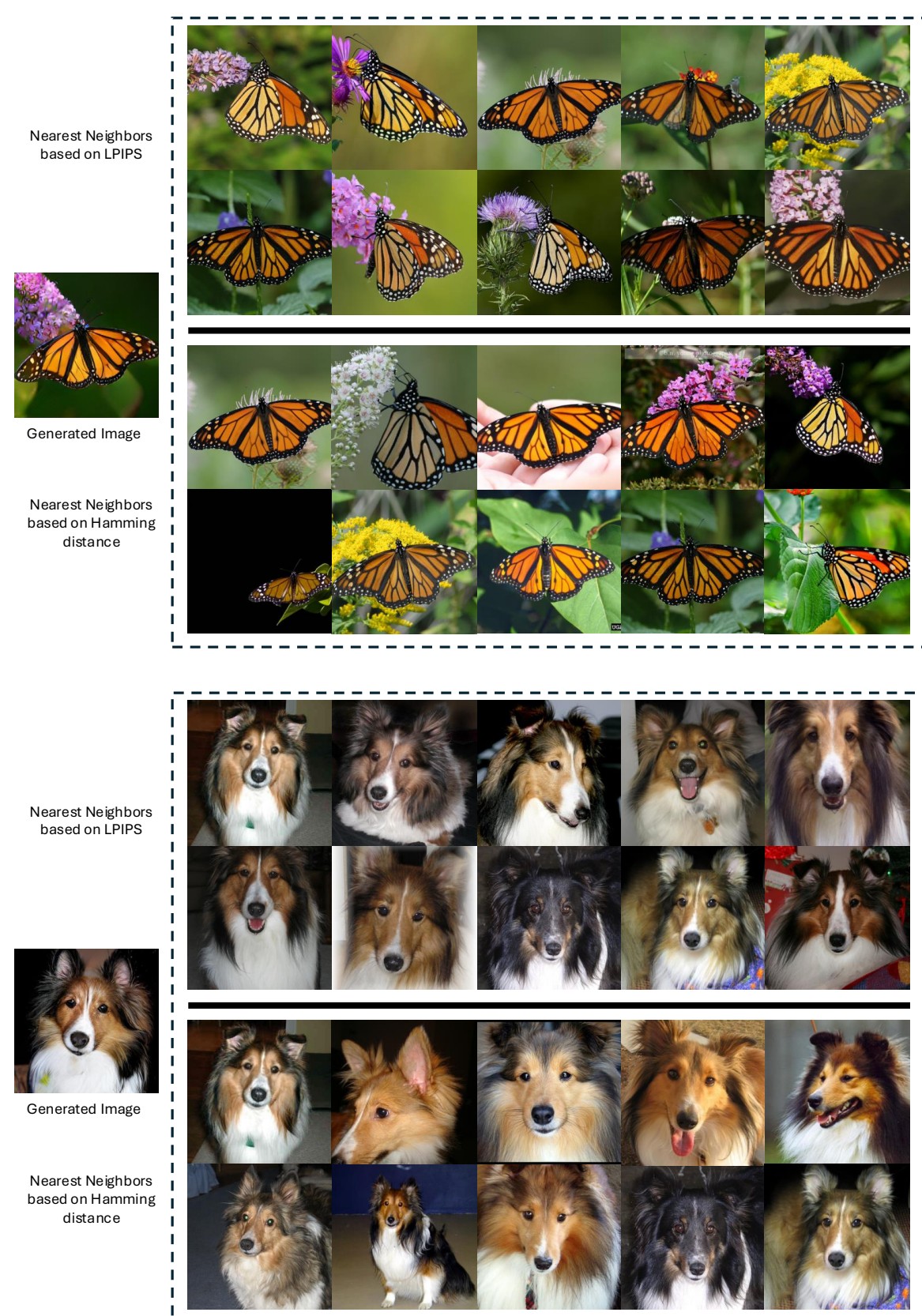

Figure 10: **Nearest Neighbors of two generated Images.** The figure shows the ten nearest neighbors in the ImageNet training set in terms of LPIPS (*top*) and Hamming distance (*bottom*).

