# OpenReview forum: "MaskBit: Embedding-free Image Generation via Bit Tokens"
_TMLR — Accepted by TMLR_

### Review · Reviewer_eg8T · 2024-10-04

**Summary Of Contributions:**

This paper introduces a novel approach to token-based class-conditional image generation through embedding-free binarized bit tokens, which the authors claim contain interesting semantic structure, unlike the more semantically opaque representations found in the lookup tables standard to other token-based approaches. The paper also examines the design space of an exemplar token-based model, VQGAN, and proposes a series of changes that lead to VQGAN+, an updated model that shows substantially better reconstruction performance relative to VQGAN, and achieves state-of-the-art performance on the ImageNet $256 \times 256$ generation benchmark. The scheme for binarizing tokens according to sign value also allows for a simple way to perform bit masking through zeroing out groups of bits, which allows for separate training of a Transformer-based, LLM-style generative image model.

**Audience:**

Yes

**Broader Impact Concerns:**

The authors have adequately addressed any potential broader impact concerns in their statement in the paper.

**Claims And Evidence:**

Yes

**Requested Changes:**

The bulleted items listed above should be fairly easy to address, and I hope that the authors discuss them in the rebuttal and consider incorporating them into the paper for the benefit of other readers.

The larger, looming question I have regarding the potential influence of ImageNet on the results is harder to conclusively address without conducting larger-scale experiments that I concede would place an unnecessary burden on the authors. As I mentioned above, I believe that the paper makes a worthy contribution even if it leaves that issue somewhat unresolved. At a minimum, I think the nearest-neighbor analysis mentioned above would help push this issue to the side somewhat. Beyond that, I am primarily interested in what is made of this issue during the rebuttal discussion with the authors and from the input from the other reviewers.

**Strengths And Weaknesses:**

### **Strengths**

My overall impression of the paper is quite positive. The problem is well motivated and is grounded in an area that I think tends to get inadequate attention in the generative modeling literature, namely the nature and design of the data representation formed in "Stage I" versus the far more commonly explored improvements to the *generative* component of the problem in "Stage II." The writing and presentation are generally strong throughout the paper, with the section discussing the development of VQGAN+ and its improvements to the baseline being especially well presented. The introduction of masking into lookup-free quantization (LFQ), while straightforward, is clean and elegant. The ImageNet results are compelling, and the results about the semantic structuring of the representations are quite interesting (although on that latter point, please see below). The authors are also doing a useful service to the research community by making their implementation publicly available.

### **Weaknesses**

The authors acknowledge various potential limitations of having trained and tested only on ImageNet, and I appreciate their rationale for not extending the work to larger-scale data sets. Nevertheless, I do have a few lingering questions about whether the customizations that resulted in the superior performance of VQGAN+ are more specifically suited to ImageNet and its characteristics (i.e. "focus on a single foreground object") or if they can be expected to provide similar benefits on image data sets with more varied structure, which are the type more likely to be leveraged in current applications. For similar reasons, I wonder whether the findings about the bit tokens' semantic structuring will also hold on more complex data sets. (I suspect so, but I can't be sure.)

This is the only area where there may be a slight mismatch between the paper's claims and the evidence provided, but I do not believe it is to the point where I would say that the authors have not satisfied the claims-and-evidence criterion. The paper's explicitly claimed contributions are delivered on, and it's only *implicit* claims (such as the implied generality from referring to VQGAN+ as "a modern VQGAN") that are perhaps not completely established ... at least, not yet. This issue aside, I think that the paper makes a useful contribution as it stands, and the authors' public release of their implementation will help settle some of these remaining questions in follow-up work.

My other concerns about the paper are relatively minor and mostly amount to areas that I believe would benefit from clarification or could otherwise be strengthened. A few examples are:
* Given the exclusive focus on ImageNet, it would be helpful to have a nearest-neighbor analysis of generative output versus training data, such as reported in [1]. There is also an interesting opportunity to conduct a neighbor analysis by comparing the LFQ encodings of data versus generated samples, as opposed to the LPIPS-based approach of [1]. This would allay some concerns about whether the optimizations presented in this work are targeting the ImageNet data specifically or can effectively generalize beyond them.
* In testing the benefits of perceptual loss, the authors mention that "incorporating intermediate features into this loss can improve reconstruction further, [but] it negatively affects generation performance." Does this mean higher FID in generation or a greater tendency to memorize the training data? In Section 2.3, every design choice is made to improve reconstruction FID, so I got curious when something (i.e. intermediate features) could have improved rFID further but was ruled out due to undesired effects on the downstream generation task. Presumably, the effects on generation of every choice made in Section 2.3 were tested, and if so, the authors might consider including those results. (There are some choices for how best to present them, and the large amount of real estate currently devoted to Figure 2's bar graph might be a good spot.)
* In Section 2.2, the authors mention that "[u]nless specified otherwise, all networks are trained with a batch size of 256 for 300,000 iterations." This seems clear enough, so am I correct in assuming that all of the listed improvements (other than longer training) were the result of training from scratch in the new configuration and not from a previous checkpoint?
* The semantic structuring results in Section 3 are interesting, but I have to confess that I remain unsure as to exactly how they were obtained. The procedure itself seems clear enough: You have 12 bits, and you flip one of them across all bit tokens and decode the results to see what happens. This gives you the 12 variations on reconstruction, such as what is shown in Figures 3 and 6, indicating that "the reconstructed images from these bit-flipped tokens are still visually and semantically similar to the original images." Here's where I got confused:
> We note that this behaviour is not present in VQGAN methods. Conducting the same experiment with VQGAN+ leads to non-meaningful output, sharing no semantic or visual similarities with the original image.

If it's not VQGAN and not VQGAN+, what's doing the decoding for reconstruction? I may be missing something obvious here, but I remain stuck on it. And since it is one of the highlighted findings of the paper and the motivation for MaskBit, I want to make sure I understand it.

[1] Esser, P., Rombach, R., and Ommer, B. (2021). Taming Transformers for High-Resolution Image Synthesis.

---

> ### Author Response · Authors · 2024-10-19
> **To Reviewer eg8T (1/2)**
>
> We thank Reviewer eg8T for the constructive and detailed review as well as all the suggestions to strengthen the paper, and we carefully address the concerns below. We appreciate the reviewer’s acknowledgement of the motivation for the study, the presentation of it, as well as the “compelling” results on ImageNet.
>
> >W1.1: Do the findings about the bit tokens' semantic structuring also hold on more complex data sets?
>
> We thank the reviewer for the constructive suggestion. While we’re not able to properly train on COCO due to the limited rebuttal time, we are happy to provide ***zero-shot*** experiments to showcase the semantic structure of bit tokens. In particular, we take the same 12-bit tokenizer model used for the bit-flipping experiment on ImageNet, and apply it to validation images of COCO. While the total reconstruction performance is somewhat worse due to the zero-shot setting, we can still observe that these structural properties of the latent space effectively transfer to the complex dataset COCO.
>
>
> We provide the requested images in [Sec. 1 of this anonymous url link](https://anonymous-maskbit.github.io/).
>
> > W1.2: Do the customizations that lead to the superior performance of VQGAN+ also provide benefits on image data sets with more varied structure?
>
> We thank the reviewer for the interesting question. Similar to the experiment presented in W1.1, we conduct ***zero-shot evaluation*** on COCO to quantitatively show how well the customizations leading to VQGAN+ work on datasets with more varied structure. To ensure a fair comparison, we use the same models as in Sec. 2: the original Taming-VQGAN model (7.96 rFID on ImageNet), the VQGAN+ (1.66 rFID on ImageNet), and its embedding-free variant (1.61 rFID on ImageNet). All these models have only been trained on ImageNet.
>
> | Model | COCO 0-shot rFID |
> |-----------|-----------|
> | Taming-VQGAN | 16.3 |
> | VQGAN+ | 8.7 |
> | Embedding-free variant | 8.3 |
>
> We make the observation that the customizations presented in this work are also beneficial on datasets with more varied structure. While the zero-shot setting reflects in the magnitude of the rFID scores, both VQGAN+ and its embedding-free variant achieve significant performance gains over the baseline. Similar to ImageNet, the embedding-free variant achieves better results compared to the plain VQGAN+. We believe that this experiment gives evidence that at least points towards a potential benefit of these customizations of more complex datasets. We hope that future work will look into the potential of VQGAN+ and its embedding-free variant on large-scale datasets such as LAION, and provide a definitive answer.
>
> > W2: Could the authors visualize the nearest neighbors of the generated images to the training set?
>
> We thank the reviewer for the suggestion. We visualize the nearest neighbors in terms of two distances: 1.) We follow the experiment design of the Taming-VQGAN paper and use the LPIPS distance, and 2.) we use the hamming distance of the corresponding latent bit tokens. The visualizations of the 10 nearest neighbors can be found in [Sec. 2 of this anonymous url link](https://anonymous-maskbit.github.io/).
>
>
> The results show that the model is not just memorizing samples from the training set, but generates indeed new images. Beyond that, we notice some differences in the 10 nearest neighbors between LPIPS and the hamming distance. For example, if we consider the neighbors to the generated images of the parrot or the space rocket, we observe that the nearest samples according to LPIPS focus more on staying true to the reference color palette, while this is less the case for the closest samples according to the hamming distance. This finding aligns with the observations in the bit-flipping experiment, where with small changes in the hamming distance, the visual attributes can vary quite a bit, while the semantic content stays the same.

---

> ### Author Response · Authors · 2024-10-19
> **To Reviewer eg8T (2/2)**
>
> > W3: In testing the benefits of perceptual loss, the authors mention that "incorporating intermediate features into this loss can improve reconstruction further, [but] it negatively affects generation performance." Does this mean higher FID in generation or a greater tendency to memorize the training data?
>
>
> We thank the reviewer for the question. In this case, it means that it hurts the generation FID. Using a perceptual loss based on logits, for example from ResNet, puts the focus on semantic similarity of the reconstructed image to the input image. When also using intermediate features as in LPIPS, the similarity is instead measured at low- and high-level feature maps. We empirically find that a logits-based loss is beneficial for the learning of the generation model. We simply shared this detail, as other researchers on image tokenizers might stumble over the same effect in their studies and could find this useful.
>
> It is noteworthy that the work [A] has studied this effect in detail independently, and has made similar observations.
>
> [A] Rethinking the Objectives of Vector-Quantized Tokenizers for Image Synthesis. In CVPR, 2024.
>
> > W4: Presumably, the effects on generation of every choice made in Section 2.3 were tested, and if so, the authors might consider including those results.
>
> As the generation model is computationally expensive to train, we were unfortunately unable to test every choice in Section 2.3; otherwise, we would be happy to include them. In this particular case, we were aware of the trade-offs between logits-based perceptual losses and “multi-feature”-based perceptual losses such as LPIPS, which has been studied by [A]. We therefore took extra care regarding this design choice. Our paper has studied the Stage-II model performance for the final VQGAN+ model and its embedding-free variant. The focus of the generation model experiments were thus more on the proposed MaskBit model with bit tokens and grouping. Still, the overall results of both tokenizers show the overall clear benefit for generation from modifications discussed in Sec. 2.3.
>
> > W5: In Section 2 [...] am I correct in assuming that all of the listed improvements (other than longer training) were the result of training from scratch in the new configuration and not from a previous checkpoint?
>
> Yes, that is absolutely correct. All experiments are trained from scratch for 300k iterations except for the experiment “longer training” and “embedding-free” (Last two experiments in Sec. 2 / Fig. 2).
>
> > W6: If it's not VQGAN and not VQGAN+, what's doing the decoding for reconstruction in the bit flipping experiment?
>
> We thank the reviewer for the question. We acknowledge that the terminology gets somewhat blurry here. We obtain two Stage-I models from our studies in the end: (1) VQGAN+ that uses the traditional learnable embedding table (1.66 rFID), and (2) its embedding-free variant that uses bit tokens (1.61 rFID). The decoder architecture used for VQGAN+ and its embedding-free variant are the same, except for the input dimension of the first layer (in order to process different latent dimensions). Therefore, the decoder of the embedding-free model is used to decode the image. We clarified this in the updated draft.

---

> > ### Comment · Reviewer_eg8T · 2024-10-21
> > **Response to Authors**
> >
> > I thank the authors for their comments. All of the points that I raised have been addressed to my satisfaction.
> >
> > I encourage the authors to include the nearest-neighbor and zero-shot COCO results in the supplement/appendix of their final revision, as I think they lend useful support to the paper's claims.
> >
> > I have no remaining concerns about the paper.

---

> > > ### Author Response · Authors · 2024-10-21
> > > **Adding the rebuttal findings in the current revision**
> > >
> > > We thank the reviewer for their quick response. We are happy to put the nearest-neighbor results as well as the zero-shot COCO results in the appendix. The change is reflected in the current version, as follows:
> > >
> > > 1. Appendix D includes COCO zero-shot bit flipping results.
> > >
> > > 2. Appendix E contains the quantitative results of zero-shot reconstruction on COCO.
> > >
> > > 3. Appendix F adds the nearest neighbor analysis.

---

### Review · Reviewer_r6hD · 2024-10-10

**Summary Of Contributions:**

* Propose group of bits masking as an extension of the framework proposed in (Yu et al., 2024), based on lookup-free quantization and masked modelling.

* Demonstrate that SOTA results can be obtained on ImageNet from lookup-free quantization + (group of bits) masked modelling.

* Provide open-source implementation and reproducible experiments of SOTA image tokenizers and masked transformers for image generation on ImageNet.

**Audience:**

Yes

**Broader Impact Concerns:**

The broader impact concerns is reasonable.

**Claims And Evidence:**

Yes

**Requested Changes:**

* To me, the most important point concerns the explanation of contributions (cf Weaknesses). How does MaskBit differ from MAGVIT-v2, which propose lookup-free quantization and masked modelling? In the abstract, it is written "The second contribution demonstrates that embedding-free image generation using bit tokens achieves a new state-of-the-art FID of 1.52 on the ImageNet 256×256 benchmark". I am completely ok with this. However, in other parts of the paper, it is not that clear anymore. For example, in the contributions page 2: "We develop a novel embedding-free generation framework, MaskBit." Again (cf Weaknesses), how is MaskBit a novel embeding-free generation framework?

**Strengths And Weaknesses:**

Strengths:

* Open-source and reproducible implementation of look-up free quantization autoencoders and masked transformers, reaching SOTA performance on ImageNet.

* Very clear, detailed, and reproducible experimental setting. The improvements on both VQGAN++ and bit-masked transformers are strong and can serve as a new baseline for practitioners.

Weaknesses:

* The distinction between MAGVIT-v2 (Yu et al.) and MaskBit is not clear from reading the paper. Indeed, it is stated in Page 6: "This motivates us to propose MaskBit for Stage-II, a transformer-based generative model that generates images directly using bit tokens, removing the need for the embedding lookup table as
required by existing methods (Chang et al., 2022; Yu et al., 2024a)". However, Yu et al. precisely propose lookup-free quantization, which does not require embedding lookup table as far as I understand. The fact that (Yu et al., 2024a) propose lookup-free quantization is even acknowledged by the authors in other parts of the paper. Thus, how can we compare both approaches? What are the differences in MaskBit?

* The main contribution, regarding the difference with MAGVIT-v2, seems to be the Group of Bits modelling. However, its impact seems weak, with only 0.12 improvement of FID evaluated on a single dataset and without any confidence interval on the measure. I doubt that the improvement given by group of bits is really robust. The slight variation of gFID could be "in the noise". I understand that launching several trainings is too costly, but at least several sampling of generated/reference images is doable and can give a gross idea of the standard deviation of gFID.

---

> ### Author Response · Authors · 2024-10-19
> **To Reviewer r6hD (1/2)**
>
> We thank Reviewer r6hD for the review, the questions, and the feedback to strengthen the paper, and we carefully address the concerns below. We are delighted that the reviewer finds the experimental setting “very clear, detailed, and reproducible”.
>
> > W1/R1: What are the differences between MAGVIT-v2 and MaskBit?
>
>
> While MAGVIT-v2 and MaskBit share similarities in the tokenizer (“Stage-I”), the generation model (“Stage-II”) of MAGVIT-v2 and MaskBit are very different.
>
> Specifically, in MAGVIT-v2, the LFQ encodings are ***only used*** in the Stage-I model, while its Stage-II generation model uses a codebook to learn new embeddings for generation. The LFQ encodings of Stage-I are used as an index into the learnable codebook of Stage-II in MAGVIT-v2. Thus, MAGVIT-v2 ***does not share any learnable representation*** between both stages and the generation model learns a new representation/embeddings from scratch.
>
> In our studies on the learned latent representations, we found that the learned bit tokens exhibit an interesting semantic structure. We refer to Sec. 3 *“Bit Tokens Are Semantically Structured”* and Fig. 3. Motivated by these findings, we investigate whether bit tokens can also be used effectively in the generator, which has not been answered by MAGVIT-v2 or any other prior work. To this end, we propose MaskBit that shares the representation, and we study the effects in detail. In appendix C of the original draft, we provided more discussion on why MAGVIT-v2 and MaskBit’s internal representation are not equivalent, due to the specific structure of bit tokens.
>
> In Table 3a, we show the effectiveness of using a shared representation, as it outperforms variants that use a codebook. In Table 1, we show that using 64 steps (as used in MAGVIT-v2), MaskBit achieves an FID of 1.62 compared to 1.78 MAGVIT-v2 while using a 50% smaller Stage-I network (54M vs. 116M). Even when increasing the codebook size to be the same as MAGVIT-v2 (i.e., 18 bits), Table 2 still shows significant improvements. We note that using 18 bits performs worse in our setting (potentially due to our much smaller Stage-I model size).
>
> > W2.1: The main contribution, regarding the difference with MAGVIT-v2, seems to be the Group of Bits modelling.
>
>
> We refer to the answer W1/R1 regarding all the differences between MAGVIT-v2 and MaskBit. In short, the Stage-II model of MAGVIT-v2 and MaskBit work conceptually and empirically quite differently. MAGVIT-v2 learns a new codebook in Stage-II, and thus does not share a latent representation between Stage-I and Stage-II. MaskBit, however, shares a latent representation between both stages by using bit tokens. On top of this idea, MaskBit then uses ***bit masking*** and ***grouping*** to improve results.
>
>
> Besides the differences in the Stage-II model, we kindly reiterate some of the main claims the paper studies. In Sec. 2, we study the effect of several modifications to the VQGAN training framework and experimentally validate the effect of these changes. All these customizations lead to a model coined VQGAN+ as well as its embedding-free variant, which is then used in the MaskBit generation framework. In Sec. 3, we study how the structure of bit tokens relates to the semantic and visual content of the decoded image.
>
> > W2.2: Can an improvement of 0.12 gFID be seen as significant or could it be “in the noise”? What is the standard deviation in gFID for MaskBit models?
>
>
> We thank the reviewer for the suggestion to report standard deviation. It is noteworthy that in the gFID range that MaskBit and concurrent work achieve, improvements of 0.12 gFID are quite significant. Indeed, if changes of 0.12 gFID would be “in the noise”, this would raise questions regarding the overall performance of MaskBit and the soundness of the comparison to prior art. We thus launch 3 evaluations of the same generation model to measure the standard deviation. In each evaluation, we follow the standard protocol and generate 50k samples. We evaluated the 12-bit 2 groups MaskBit (64 sampling steps) and the 14-bit 2 groups MaskBit (64 sampling steps) which have a standard deviation of 0.011 and 0.009, respectively.
>
> ***Given the low standard deviation, we think the improvements reported in the paper are reliable and significant.*** Since ***none*** of the prior works report the standard deviation of their models, we cannot provide a comparison of the standard deviation with prior works.

---

> > ### Comment · Reviewer_r6hD · 2024-10-21
> > **Response**
> >
> > I thank the authors for the clear answer.
> >
> > The difference between MAGVIT-v2 and the proposed method is now clearer. I understand that MaskBIT is directly using the embeddings from the Stage-1 VQGAN. A necessary ablation study is to understand the impact of using directly LFQ-embeddings vs using index and learning new embeddings. Could the authors confirm that this ablation study correspond to Table 3 (a), second row (tokenizer embedding-free but no generator embedding-free)? Thus, the setting of row 2 corresponds exactly to the setting of MAGVIT-v2?
> >
> > As for the other concerns that I raised, I am satisfied by the authors' response.

---

> > > ### Author Response · Authors · 2024-10-21
> > > **Table 3a experiment setting**
> > >
> > > > Does the setting of row 2 in Table 3a (tokenizer embedding-free but no generator embedding-free) correspond exactly to the setting of MAGVIT-v2?
> > >
> > > We thank the reviewer for their quick response. Indeed, the setting of an embedding-free tokenizer and a generator with a learnable codebook corresponds to the setting of MAGVIT-v2.
> > >
> > > For completeness, we would like to mention that there are several differences in the details of the Stage-I model between MAGVIT-v2 and our tokenizer, regarding the training recipe, tokenizer size, and codebook size, which is why we also provide a direct comparison to MAGVIT-v2’s final scores in Table 1 and Table 2.

---

> ### Author Response · Authors · 2024-10-19
> **To Reviewer r6hD (2/2)**
>
> > R2: How is MaskBit a novel embedding-free generation framework?
>
> The proposed MaskBit framework provides novelty in both Stage-I and in Stage-II.
>
> For Stage-I, the paper studies many changes to the tokenizer, leading to two new models: VQGAN+ as well as its embedding-free variant. We are not aware of any prior work that has studied the design and training recipe of image tokenizers in such a detailed and sound study. As tokenizer models are the foundation for generative models working in latent space, we believe that there exists an audience interested in this systematic study. Specifically, in this study, the paper experiments with ideas proposed by prior art (e.g., removal of attention layers), ideas not transparently discussed in prior art (e.g., perceptual loss) and novel architectural modifications such as the redesign of the PatchGAN discriminator to align with the encoder (see the paragraph ***Discriminator Update***).
>
> For Stage-II, MaskBit is the first framework that uses a shared representation between Stage-I and Stage-II (see also our reply to *W1/R1: Discussion of differences between MAGVIT-v2 and MaskBit*). On top of this, we design a masking scheme specific to bit tokens and propose bitwise grouping, leading to state-of-the-art performance on the competitive class-conditional image generation benchmark.

---

### Review · Reviewer_UHqj · 2024-10-14

**Summary Of Contributions:**

This paper presents VQGAN+, an improved version of the VQGAN model through step-by-step examination of VQGANs, which not only modernizes the existing VQGANs but also enhances transparency and reproducibility. It also introduces MaskBit, which utilizes an embedding-free mechanism based on bit tokens. It achieves state-of-the-art results on ImageNet with fewer parameters, demonstrating the effectiveness of the bit token representation.

**Audience:**

Yes

**Claims And Evidence:**

No

**Requested Changes:**

- I strongly recommend that the authors provide the code, at least to the ACs and reviewers, during the discussion period, as I believe it is the most significant contribution of this paper.

- The authors should demonstrate that VQ-GAN is not robust against latent manipulation. However, it may be challenging to make a fair comparison between bit latents and VQ latents, as changing a code of VQ latent to a random one affects the latent representation more significantly than simply flipping a bit in MaskBit's latent.

- I suggest that the authors explore bit latents trained on datasets other than ImageNet to verify whether the properties of bit tokens are universal.

- I suggest that the authors include illustrations, pseudo-code, or mathematical formulas for the training and inference processes of MaskBit.

- Figure 4 could be more informative by including notations and terminology used throughout this paper.

- The phrase "14 Bits are Enough for ImageNet" may be misleading and could confuse readers. A more precise expression is recommended.

**Strengths And Weaknesses:**

**Strength**

- Section 2 effectively outlines the step-by-step improvements to VQ-GAN, making it easier to understand the impact of each modification.

- They plan to release the code for all experiments, which will be valuable for the deep generative modeling community, as there are few works that publish reproducible code.

- The empirical results, especially the FID score achieved by MaskBit, are impressive and set a new benchmark for the field.

**Weaknesses**

- Although making the code available to reproduce the experiments is a key contribution of this work, the reviewers currently do not have access to it.

- Regarding Figure 3, readers may not be convinced that this robustness property occurs exclusively with bit-token latents. The authors simply state, 'We note that this behavior is not present in VQGAN methods.' Even if that is true, it remains unclear why only bit tokens exhibit robustness against latent perturbations.

- The discussion on the semantic structure of bit tokens from a robustness perspective is not well connected to their ability to provide a good representation for generation. Moreover, the current experiments do not clearly demonstrate how this affects generation performance.

- The inference procedure of MaskBIT is not clearly explained.

- It is unclear from the experiments how much the bit-token latents and bit grouping each contribute to MaskBit's improvement over MaskGIT.

---

> ### Author Response · Authors · 2024-10-19
> **To Reviewer UHqj (1/4)**
>
> We thank reviewer UHqj for the review and feedback to strengthen the paper, and we carefully address the concerns below. We appreciate the reviewer’s acknowledgement of the “impressive” results of MaskBit.
>
> > W1/R1: I strongly recommend that the authors provide the code during the discussion period.
>
> We are deeply committed to the code release. Naturally, large codebases require clean-up, more testing and proper documentation before public release, which is what we are working on at the moment. Nevertheless, during the review period, we are happy to share the code snippets for Stage-I model, Stage-II model, and the inference:
>
> 1. Stage-I Model / Quantizer: https://paste.rowin.dev/jyYOUk3M
> 2. Stage-II Model: https://paste.rowin.dev/nF3wUrWC
> 3. Inference: https://paste.rowin.dev/tDtjLCyX.
>
> > W2/R2: demonstrate that VQ-GAN is not robust against latent manipulation. However, it may be challenging to make a fair comparison ...
>
> We thank the reviewer for the comment. We are happy to provide the visual example of applying corruptions to VQ latents generated by VQGAN. For this experiment, we follow the same setup as the bit-flipping experiment presented in the paper. We present the results in [Sec. 3 of this anonymous url link](https://anonymous-maskbit.github.io/).
>
> We kindly emphasize that the purpose of this bit-flipping experiment was to show how the inherent structure of bit tokens relates to the content (see Fig. 3 and Fig. 6 in the paper), but ***not to make assessments about adversarial robustness***. We simply choose the bit-flipping experiment to show that relationship between structure and semantics. We removed the wording “robustness” in the manuscript to avoid confusion in the updated draft. The observation made for this study is that there is a strong connection between the structure and the semantic representation of bit tokens. Flipping bits in bit tokens does not change the semantic content of an image, but significantly changes appearance-related attributes (texture, exposure, colors, etc.). As VQGANs do not have this kind of structure, we agree with the reviewer that it is hard to “make a fair comparison” and that is why we did not show VQGAN analogy in the paper. Moreover, we would like to emphasize that, unlike MaskBit, ***all prior work does not share a representation between Stage-I (tokenizer) and Stage-II (generator)***. Thus, the two takeaways and claims, which we find supported by our analysis are:
>
> 1. Bit tokens have a unique structure by design.
> 2. This structure leads to a structured semantic representation.
>
> It is noteworthy that this structure affects how bit tokens are projected into a higher dimensional space used by the generation model internally. In the original draft, we provided an example of this in appendix C.

---

> ### Author Response · Authors · 2024-10-19
> **To Reviewer UHqj (2/4)**
>
> > W3: The discussion on the semantic structure of bit tokens from a robustness perspective is not well connected to their ability to provide a good representation for generation. Moreover, the current experiments do not clearly demonstrate how this affects generation performance.
>
> We thank the reviewer for the question. We attempt to address it in two steps, as detailed below.
>
>
> ***1. Connection of semantic structure of tokens and generation:***
>
>
> We know from prior work [A] that a strong semantic representation in the latent space is important for good generation performance. Specifically, the prior work [A] has made the following observation: *Semantic compression within VQ tokenizers benefits the generative transformer.* This observation was made by using different variations of the LPIPS perceptual loss. For details, we refer to the discussion in [A]. Moreover, this can also be supported by the findings in our experiments:
>
>
> *The standard perceptual loss is LPIPS, which uses VGG’s intermediate activations to compute its score/loss. Replacing it by a ResNet logits-based perceptual loss works well for generation. (see Sec. 2.3, Perceptual Loss).*
>
>
> Bit tokens exhibit this uniquely constrained structure, which leads to a structured semantic representation. In appendix C, we discussed how this structured representation affects the internal generation model feature representation per tokens. In particular, bit tokens can be seen as corners of an n-dimensional hyper-cube, while learnable embeddings can be somewhat arbitrarily distributed in space. ***Hence, it is this structure that is the key difference between bit tokens and learnable codebooks.***
>
>
> We have studied this structure in more detail, and discussed the findings in Sec. 3 *“Bit Tokens Are Semantically Structured”*. ***Our claim that the structure in bit tokens leads to a structured semantic representation is supported by our findings in Fig. 3 and Fig. 6.***
>
>
> [A] Rethinking the Objectives of Vector-Quantized Tokenizers for Image Synthesis. In CVPR, 2024.
>
>
> ***2. How the use of bit tokens affects generation performance:***
>
>
> In the paper, we demonstrated with various experiments regarding how the use of bit tokens in the generation stage yields better performance than learning embeddings. For example, we compare MaskBit to MAGVIT-v2, as MAGVIT-v2 also uses an embedding-free tokenizer but ***learns new embeddings for the generation model*** (“Stage-II”), while MaskBit uses bit tokens in the generation model.
>
>
> In Table 1 of the paper, we provided a direct comparison between MaskBit and MAGVIT-v2, demonstrating improvements of 0.16 gFID (1.62 vs. 1.78) when using bit tokens vs. not using bit tokens in Stage-II with 64 sampling steps. This improvement is especially significant, considering the fact that MaskBit uses a much smaller Stage-I model (54M vs. 116M).
>
>
> In Table 2 of the paper, we compared MAGVIT-v2 and MaskBit under the best setting of MAGVIT-v2 (codebook size 262144, 64 sampling steps, etc.), which, however, is not the best setting for MaskBit. Nevertheless, the scores still demonstrate improvements of using bit tokens in Stage-II (improvements of 0.11).
>
>
> In Table 3a of the paper, we compared two settings using the same (our) tokenizer. The 2nd row does not use bit tokens for generations, the 3rd row does. The score improves by 0.13 (1.95 vs. 1.82), showing the advantages of using embedding-free generators.
>
>
> ***We believe we have provided sufficient evidence supporting the potential of bit tokens to increase generation performance.***
>
>
> > W4: The inference procedure of MaskBIT is not clearly explained.
>
>
> We make no changes to the inference procedure compared to prior art (MaskGit, MAGVIT, MDT), except for the support of sampling tokens in groups independently. Appendix B.3 lists the methods from prior art that were used for inference. As the code for sampling is short, we are happy to provide a relatively clean and anonymized code snippet [at this anonymous url link](https://paste.rowin.dev/tDtjLCyX). Finally, we would like to emphasize that, as promised, we are working on cleaning and verifying the code (both training and inference, and checkpoints) for open-sourcing, allowing the community to directly work with all our results.

---

> ### Author Response · Authors · 2024-10-19
> **To Reviewer UHqj (3/4)**
>
> > W5: How much do the bit-token latents and bit grouping each contribute to MaskBit's improvement over MaskGIT.
>
>
> In Table 3 of the paper, we provided ablations of these contributions. In each experiment, to ensure fairness, we varied only a single factor, while fixing all the others. For quick reference, we reiterate the relevant results here.
>
>
> | Framework | Groups | gFID |
> |-----------|-----------|-----------|
> | MaskGIT (official) | 1 | 4.02 |
> | MaskGIT w./ VQGAN+ | 1 | 2.12 |
> | MaskBit | 1 | 1.95 |
> | MaskBit | 2 | 1.82 |
>
> The first row corresponds to the official MaskGIT scores, establishing the baseline. The second row corresponds to MaskGIT with our proposed VQGAN+, already achieving a great performance of 2.12. The 3rd row shows the contribution of bit token latents with a single group, which is an improvement of 0.17 gFID. The last row shows the contribution of enabling bit grouping with two groups, leading to further improvements of 0.13 gFID.
>
>
> > R3: I suggest that the authors explore bit latents trained on datasets other than ImageNet to verify whether the properties of bit tokens are universal.
>
>
> We thank the reviewer for the suggestion. We note that the paper does not make the claim that the properties of bit tokens are “universal”. We show the bit token properties on the ***standard and most-widely used*** class-conditional image generation benchmark, i.e., ImageNet. We believe that the experimental evidence supports the claim of a structured semantic representation on ImageNet.
>
>
> Nevertheless, we are happy to provide ***zero-shot*** results on the more complex dataset COCO. Due to the rebuttal time, we are unable to properly train on COCO, but believe the zero-shot results are a solid indicator. For a fair comparison, we used the same models as in the Sec. 2 experiments for ImageNet and evaluated them directly on the COCO validation set:
>
>
> | Model | COCO 0-shot rFID |
> |-----------|-----------|
> | Taming-VQGAN | 16.3 |
> | VQGAN+ | 8.7 |
> | Embedding-free variant | 8.3 |
>
> While the magnitude of rFID scores reflect the zero-shot nature of this experiment, the relative difference between these models support our findings regarding the benefit of VQGAN+ and its embedding-free variant. Moreover, we also repeat the bit-flipping experiment on COCO in zero-shot fashion. The results can be found in [Sec. 1 of this anonymous url link](https://anonymous-maskbit.github.io/). For this, we take the same 12-bit tokenizer model used for the bit-flipping experiment on ImageNet, and apply it to validation images of COCO. While the total reconstruction performance is somewhat worse due to the zero-shot setting, we can still observe that the structural properties of the latent space effectively transfer to the complex dataset COCO.
>
>
> > R4: I suggest that the authors include illustrations, pseudo-code, or mathematical formulas for the training and inference processes of MaskBit.
>
>
> We thank the reviewer for the suggestion. We are happy to provide pseudo-code for training and inference of MaskBit. It is noteworthy that the general training and inference framework follows prior arts in masked image generation (MaskGIT, MAGVIT, MAGVITv2). As promised, we will provide the full code online for everyone to use, which we believe might be more informative to practitioners.
>
>
> Pseudo inference code:
>
>
> ```
> function sampling -> Image:
>
> tokenizer <- The Stage-I model.
> generator <- The Stage-II model.
> num_bits <- The number of bits in a bit token.
> num_groups <- The number of groups in a bit token.
> num_steps <- The number of sampling steps.
> tokens <- Completely masked set of tokens.
>
> for i in range(num_steps):
>     logits <- infer_logits_with_cfg(generator, tokens)   # classifier-free guidance
>     predicted_tokens <- sample_from_logits(logits)
>
>     num_tokens_to_keep <- get_num_tokens_to_keep(i)
>     tokens_to_keep <- sample_num_tokens_with_confidence(predicted_tokens, num_tokens_to_keep, logits)
>     tokens <- merge(tokens, tokens_to_keep)
>
> tokens <- combine_bit_groups(tokens, num_bits, num_groups)
> image <- decode_tokens_to_image(tokenizer, tokens)
>
> return image
> ```
>
> Pseudo training code:
>
> ```
> function train_maskbit -> Model:
>
> tokenizer <- The Stage-I model.
> generator <- The Stage-II model.
> dataloader <- The dataloader for e.g. ImageNet.
> num_bits <- The number of bits in a bit token.
> num_groups <- The number of groups in a bit token.
>
> for image_batch, class_label_batch in dataloader:
> 	bit_tokens <- encode_images_to_tokens(tokenizer, image_batch)
> 	split_bit_tokens <- split_bit_tokens_into_groups(bit_tokens, num_bits, num_groups)
>
> 	masked_bit_tokens <- randomly_mask_tokens(split_bit_tokens)
> 	class_label_batch <- randomly_drop_class_label(class_label_batch)
>
> 	logits <- forward(generator, masked_bit_tokens, class_label_batch)
> 	loss <- cross_entropy_on_masked_tokens(masked_bit_tokens, logits)
>
> 	backward(generator, loss)
>
> return generator
> ```

---

> ### Author Response · Authors · 2024-10-19
> **To Reviewer UHqj (4/4)**
>
> > R5: Figure 4 could be more informative by including notations and terminology used throughout this paper.
>
>
> We are happy to take suggestions on which terminology/notation is missing that could make it more informative without making it too cluttered and convoluted. The purpose of Fig. 4 is to give a high-level overview of the generation framework and contrast the proposed method to commonly used VQGAN-based approaches. We therefore limit it in the original draft to showing the difference of codebooks and bit tokens, the masking and grouping.
>
> > R6: The phrase "14 Bits are Enough for ImageNet" may be misleading and could confuse readers. A more precise expression is recommended.
>
>
> We thank the reviewer for the suggestion. We have updated the expression to be ***MaskBit with 14 Bits Yields Better Performance***, which objectively reflects and summarizes our findings in Tab. 4a.

---

> > ### Comment · Reviewer_UHqj · 2024-10-22
> > **Reviewer reply**
> >
> > I thank the authors for their detailed explanations and the additional empirical results, which have addressed most of my initial concerns. In particular, I'm pleased to hear that they are working hard to make their code publicly available. I have a few follow-up questions and requests in response to their comments, as outlined below.
> >
> > > we agree with the reviewer that it is hard to “make a fair comparison” and that is why we did not show VQGAN analogy in the paper.
> >
> > I am curious about the accuracy of the statement, "We note that this behaviour is not present in VQGAN methods that learn codebooks." As the authors may acknowledge, comparing bit-latent flipping with replacing a code in VQGAN with a random code is not a fair comparison. I wonder how the decoded results would look when a code is replaced with a nearby (or nearest) code in a typical VQ-GAN, as this might present a more comparable scenario to the bit-latent case.
> >
> > > We thank the reviewer for the question. We attempt to address it in two steps, as detailed below.
> >
> > The authors' response to my initial concern (W3) is reasonable and helps readers understand the intuition behind it. I recommend incorporating this discussion into the manuscript, either in the main body or the appendix.
> >
> > > We note that the paper does not make the claim that the properties of bit tokens are “universal”.
> >
> > I agree that the paper does not need to establish the universality of this property. However, I suggest including the new experimental results, or other additional results on different datasets (e.g., FFHQ or LSUN is sufficient), within the paper. While ImageNet is a standard and widely used dataset, the authors' method for image generation should not be confined solely to it.
> >
> > > We are happy to take suggestions on which terminology/notation is missing that could make it more informative without making it too cluttered and convoluted.
> >
> > I understand that Figure 4 is intended to provide a high-level overview. However, I found the use of "bits" (what the unit of groups indicates, particularly when the number of groups is set to 1) somewhat unclear when I first read the paper. I recommend clarifying this point. It might also help readers if you explain the concept of "groups" through illustrations.
> >
> > > We have updated the expression to be MaskBit with 14 Bits Yields Better Performance...
> >
> > This revised phrasing, as proposed, and with reference to Table 4a, avoids confusion. My initial concern with the original sentence was that it seemed to suggest each ImageNet sample could be represented by just a 14-bit sequence, which is, of course, not the case.

---

> > > ### Author Response · Authors · 2024-10-24
> > > **Response to "reviewer reply" (1/2)**
> > >
> > > >I am curious about the accuracy of the statement, "We note that this behaviour is not present in VQGAN methods that learn codebooks." As the authors may acknowledge, comparing bit-latent flipping with replacing a code in VQGAN with a random code is not a fair comparison. I wonder how the decoded results would look when a code is replaced with a nearby (or nearest) code in a typical VQ-GAN, as this might present a more comparable scenario to the bit-latent case.
> > >
> > > We thank the reviewer for the comments, and we carefully address them from three perspectives:
> > >
> > > 1. Why there is no analogous experiment to bit-flipping in continuous latent spaces of VQGANs.
> > > 2. What the purpose of the bit-flipping experiment in the paper is.
> > > 3. What actions we take to strengthen the paper based on the reviewer’s feedback.
> > >
> > > ***1. Why there is no analogous experiment to bit-flipping in continuous latent spaces of VQGAN.***
> > >
> > > In the case of bit tokens, they are ***by design*** a structured space of latents, while in VQGAN the codebook entries can be distributed somewhat arbitrarily in space. Moreover, we believe this discussion needs to include the first linear layer projection for a full picture on the differences as explained in Appendix Sec. D.2.
> > >
> > > Mathematically, the space of bit tokens can be described as corners of a hypercube, while VQGAN codebook entries do not have such a structure. The hypercube structure leads to an important constraint in the generator and decoder when projecting the token to a higher dimension. We have provided discussion about this effect in Appendix Sec. D.2 (since the first version of the draft). In general, when using bit tokens with coefficients +/- 1, we can view the weights of the first projection layer as high-dimensional “basis vectors”. It is noteworthy that ***each basis vector is weighted with a +1 or a -1 coefficient before being summed together for the output projection and therefore each basis vector always contributes***. For example, consider the case of 12 bits. We obtain 12 high dimensional basis vectors for this projection. When “flipping a bit”, this means we traverse from one corner to another (connected) corner. This therefore changes the “contribution” of a basis vector from +1 to -1 or vice versa, ***having a significant effect*** on the output. Due to this structure, the 12 samples obtained in the bit flipping experiment for bit tokens are ***equidistant*** from the reference token in terms of Hamming distance.
> > >
> > > In contrast, considering a VQGAN space with learnable embeddings, the latent space is high dimensional (256 dimensions) and has no additional constraints to its structure. Picking a nearby or nearest code based on L2 distance will intuitively lead to small changes in the code compared to the original code. Thus, the significance of change in a nearest neighbor-like setting is still quite different to a change by bit-flipping for bit-tokens. Furthermore, if we take the k closest codes according to the L2 distance, the obtained codes are ***not equidistant*** and also across different samples there is no constraint on how close or far nearby tokens will be. By design, the latent spaces between embedding-free tokenizers and VQGANs look quite different, and thus it is hard to find a fair analogy.
> > >
> > > Additionally, we note that prior work does not use a shared representation between Stage-I and Stage-II models, i.e., the learned codebook is different between tokenizer and generator.
> > >
> > > ***2. What the purpose of the bit-flipping experiment in the paper is.***
> > >
> > > We would like to emphasize that the ***purpose of this experiment was merely to show the connection of the specific latent space structure of bit tokens and the image content***. Fig. 3 shows that bit tokens exhibit a structured semantic representation, which motivates us to propose MaskBit. We kindly point towards the [previous reply](https://openreview.net/forum?id=NYe2JuN3v3&noteId=YKmpe5rBM3 ) regarding the connection of this structure and the ability for bit tokens to serve as the generator’s representation. In short, we point to findings of prior studies that established that “semantic compression benefits the generative transformer”. By studying the connection between the structure of bit tokens and the image content, we show a structured semantic representation in bit tokens. Motivated by these findings, the proposed generator makes direct use of bit tokens, which we empirically show to outperform all prior arts which all use a separate learnable codebook in the generator.

---

> > > ### Author Response · Authors · 2024-10-24
> > > **Response to "reviewer reply" (2/2)**
> > >
> > > ***3. What actions we take to strengthen the paper based on the reviewer’s feedback.***
> > >
> > > Considering the different perspectives on finding analogous experimental settings between the latent spaces, we propose to address these concerns by modifying the draft.
> > >
> > > Based on the reviewer’s earlier response, we have already removed any wording related to “robustness” and therefore any explicit or implicit claims connected to this wording. Moreover, we have made the following changes to the draft, which are visible in the current revision:
> > >
> > > - *“We note that this behaviour is not present in VQGAN methods that learn codebooks.”* is changed to ***“We note that the structural differences in the latent space between bit tokens and learnable embeddings also have implications on the generator’s representation, which we describe in Appendix D.”***
> > > - The following sentence is removed: *“Conducting the same experiment with VQGAN+ using learnable embeddings leads to non-meaningful output, sharing no semantic or visual similarities with the original image.”*
> > >
> > > We believe these changes are in line with [TMLR guidelines of acceptance criteria](https://jmlr.org/tmlr/acceptance-criteria.html) that state “[running experiments] is not the only way to address such concerns. Another is simply for the authors to adjust (reduce) their claims.“
> > >
> > > The remaining two claims in this section are thus:
> > >
> > > 1. Bit tokens learn a structured semantic representation.
> > > 2. Bit tokens can effectively be used as shared representation for the generator.
> > >
> > > We think both claims are sufficiently supported by the evidence presented in the paper.
> > >
> > > > The authors' response to my initial concern (W3) is reasonable and helps readers understand the intuition behind it. I recommend incorporating this discussion into the manuscript, either in the main body or the appendix.
> > >
> > > We thank the reviewer and are happy to add this discussion to the Appendix Sec. D.1. The changes are visible in the current revision.
> > >
> > > > I agree that the paper does not need to establish the universality of this property. However, I suggest including the new experimental results, or other additional results on different datasets (e.g., FFHQ or LSUN is sufficient), within the paper. While ImageNet is a standard and widely used dataset, the authors' method for image generation should not be confined solely to it.
> > >
> > > We thank the reviewer for agreeing that ***the paper does not need to establish the universality of this property*** and that ***ImageNet is a standard and widely used dataset***. Due to the limited rebuttal time, it is unfortunately infeasible for us to properly train the Stage-I tokenizer and Stage-II generator on other datasets. Thus, we opt for the first suggestion of “including the new experimental results” in the paper. We are happy to report that we have already included the zero-shot results on COCO in Appendix Sec. E and Sec. F in the current revision of the paper.
> > >
> > >
> > > >I understand that Figure 4 is intended to provide a high-level overview. However, I found the use of "bits" (what the unit of groups indicates, particularly when the number of groups is set to 1) somewhat unclear when I first read the paper. I recommend clarifying this point. It might also help readers if you explain the concept of "groups" through illustrations.
> > >
> > > We thank the reviewer for providing clarifications regarding their suggestion to improve the explanation of grouping. We believe that it is easier to explain the concept of groups and masking in an additional illustration. Therefore, we have included the additional illustration in the Appendix Sec. C and referenced it in the main paper in Fig. 4 caption and the section about masking of bit tokens (paragraph **Masked ``Groups of Bits’’ Modeling** in Sec. 3). The changes are already included in the current revision of the paper.

---

> > > > ### Comment · Reviewer_UHqj · 2024-10-28
> > > > **Reviewer reply**
> > > >
> > > > I appreciate the authors' efforts to address my concerns. The new revisions work well, and I have no further concerns.

---

### Decision · Action_Editor_g4Px · 2024-11-26

**Recommendation:** Accept as is

**Comment:**

The paper is recognized for clear motivation, strong writing, and significant contributions to both token-based image generation and latent representation design.
Public release of code and reproducible experiments is praised, addressing community needs.
Performance results on ImageNet are compelling, establishing new benchmarks.

Key Weaknesses and Authours' rebuttal:

1) Limited Dataset Scope: Reviewers noted that experiments are limited to ImageNet, raising questions about generalizability to datasets with more complex structures (e.g., COCO or FFHQ).
Authors provided zero-shot evaluations on COCO, demonstrating some transferability.
2) Distinction from MAGVIT-v2: Reviewers initially found the differences between MaskBit and MAGVIT-v2 unclear.
Authors clarified the unique aspects, such as MaskBit's shared representations and masking scheme.
3)Robustness of Improvements: A 0.12 improvement in gFID was deemed marginal without confidence intervals.
The authors addressed this with additional evaluations showing low standard deviations, confirming the reliability of improvements.
4) Additional Visual Analysis: Suggestions included nearest-neighbor analysis of generated images to training data.
The authors added these results, showing non-memorization of training samples.
5) Terminological and Procedural Clarifications: Requests were made for clearer explanations of semantic structure, decoding procedures, and grouped bit modeling.
Authors updated sections and illustrations for clarity.

Most reviewers leaned towards acceptance, highlighting reproducibility, community benefit, and solid contributions. The AE suggested the acceptance.

**Audience:**

This paper is interested to AI researchers in general.

**Claims And Evidence:**

The paper introduces MaskBit, an embedding-free framework for class-conditional image generation using bit tokens. It presents two main contributions:

1) VQGAN+: A modernized and reproducible version of VQGAN, offering improved transparency and matching state-of-the-art performance.
2) MaskBit Framework: An embedding-free generative model operating directly on binary bit tokens. This method achieves a state-of-the-art FID score of 1.52 on ImageNet with a compact 305M parameter model.

Key Evidence Supporting Claims:
1) MaskBit achieves significant improvements over MAGVIT-v2, demonstrating the effectiveness of using bit tokens as a shared representation between the tokenizer (Stage I) and generator (Stage II).
2) The experiments show structured semantic representation in bit tokens, enabling robustness against latent perturbations.
3) The empirical results validate claims of enhanced performance and transparency with detailed ablations and architectural comparisons.